# Slow-moving rock glaciers in marginal periglacial environment of Southern Carpathians

Alexandru Onaca[1,2], Flavius Sîrbu[2], Valentin Poncoș[3], Christin Hilbich[4], Tazio Strozzi[5], Petru Urdea[1,2], Răzvan Popescu[6], Oana Berzescu[2], Bernd Etzelmüller[7], Alfred Vespremeanu-Stroe[6], Mirela Vasile[8], Delia Teleagă[3], Dan Birtaș[3], Iosif Lopătiță[1], Simon Filhol[7], Alexandru Hegyi[2,7], Florina Ardelean[1]

[1]Department of Geography, West University of Timișoara, Timișoara, Romania
[2]Institute for Advanced Environmental Research, West University of Timișoara, Timișoara, Romania
[3]Terrasigna, Bucharest, Romania
[4]Department of Geosciences, University of Fribourg, Fribourg, Switzerland
[5]Gamma Remote Sensing, Gümligen, Switzerland
[6]Faculty of Geography, University of Bucharest, Bucharest, Romania
[7]Department of Geosciences, University of Oslo, Oslo, Norway
[8]Division of Earth, Environmental and Life Sciences, University of Bucharest Research Institute, Bucharest, Romania

*Correspondence to*: Flavius Sîrbu (flavius.sirbu@e-uvt.ro)

**Abstract.** Rock glaciers, composed of debris and ice, are widely distributed across cold mountain regions worldwide. Although research on rock glaciers is gaining momentum, the distinct behaviour of rock glaciers in the marginal periglacial environments remains poorly understood. In this study, we combine remote sensing and in situ methods to gain insights into the characteristics of transitional rock glaciers in the Carpathian Mountains. We applied Persistent Scatterer Interferometry (PSInSAR) to Sentinel-1 images from 2015 to 2020 to identify areas with slope movements associated with rock glaciers and differential GNSS measurements (2019-2021) to detect the horizontal movement of 25 survey markers. Continuous ground temperature monitoring and measurements of the bottom temperature of the winter snow cover were used to examine the energy exchange fluxes characteristics of transitional rock glaciers in the Carpathians. The subsurface of one transitional rock glacier was investigated using geophysical measurements (electrical resistivity tomography and refraction seismic tomography), while petrophysical joint inversion was used to quantify the ice content. The PSInSAR methodology identified 92 moving areas (MAs) with low displacement rates (< 5 cm yr$^{-1}$). These MAs were generally located between 2000 and 2300 meters where solar radiation was minimal. Near-surface thermal measurements on four rock glaciers indicate favorable conditions for permafrost persistence, largely driven by internal ventilation processes (e.g., advection heat fluxes) throughout the winter. Very low ground surface temperatures were detected by BTS measurements over much of the investigated rock glaciers, particularly in their upper parts and within the MAs. Geophysical investigations reveal remnants of ice-poor permafrost within the Galeșu rock glacier, while petrophysical joint inversion modelling indicates a low ground ice content (~ 18 %) in its upper sector. At this site, the recorded surface displacements are more likely the result of ice-melt-induced

subsidence, solifluction, or the tilting and sliding of blocks within the active layer. The flow direction of dGNSS markers at two rock glaciers indicated consistent movement toward their fronts, a pattern typical of permafrost creep. Regarding activity status, the majority of rock glaciers in the Retezat Mountains were categorized as relict, with only 21% classified as transitional. Compared to relict rock glaciers, transitional ones are situated at a median elevation 150 m higher and have a slightly smaller median size.

## 1 Introduction

Rock glaciers are prominent landforms in the periglacial environment, serving as indicators of permafrost presence at the time of their formation (Haeberli et al., 2006). Generated through past or present permafrost creep, they are debris-dominated features typically identified by their front, lateral margins and occasionally ridge-and-furrow surface patterns (RGIK, 2023a). The geomorphic imprint of permafrost creep is often preserved even after the ice within the rock glacier has completely melted (Kellerer-Pirklbauer et al., 2022). In the Southern Carpathians, rock glaciers mostly present as relict landforms, yet retain isolated permafrost in certain areas (Vespremeanu-Stroe et al., 2012; Onaca et al., 2013, 2015; Popescu et al., 2015, 2024). Indicators such as extensive lichen cover, vegetated fronts and the overall morphological stability of many landforms suggest that permafrost creep is significantly reduced compared to the colder climatic conditions of the pre-Holocene (Popescu et al., 2017). These rock glaciers are predominantly mantled by angular, coarse-grained blocks which facilitate ground cooling (Onaca et al., 2017a). The thermal offset associated with this blocky surface layer contributes to the maintenance of subzero temperatures in the subsurface over prolonged periods (Kellerer-Pirklbauer, 2019), thereby enhancing permafrost preservation even at relatively low altitudes (Colucci et al., 2019). In addition, the `chimney effect` - an advective heat flux process (Delaloye and Lambiel, 2005) - significantly contributes to surface cooling in highly porous, openwork structures.

Permafrost creep encompasses both the internal deformation of ice within the frozen material and shearing at discrete planes within or just beneath the frozen structure (Cicoira et al., 2021). Surface displacement can also result from processes occurring within the active layer, such as ice-melt-induced subsidence, solifluction, or the tilting and sliding of blocks, which may act independently of or in addition to permafrost creep (Serrano et al., 2010; Cicoira et al., 2021). The surface kinematics of rock glaciers had garnered significant interest from the international community in recent years (Kellerer-Pirklbauer et al., 2024; Kääb and Røste, 2024; Pellet et al., 2024; Hu et al., 2025) due to the growing need to better understand how mountain permafrost responds to ongoing climate change. While the response of rock glaciers to present-day air temperature rising is intricate in many instances, increased rock glacier velocities has been linked to warmer climate (Wirz et al., 2016; Cicoira et al., 2019; Kenner et al., 2019; Kääb et al., 2021; Marcer et al., 2021; Kellerer-Pirklbauer et al., 2024). Rising temperatures within frozen debris enhance movement rates, as warming reduces the viscosity of the ice and promotes additional lubrication from infiltrating water (Kääb et al., 2007). According to Necsoiu et al. (2016), slow-moving rock glaciers in the Southern Carpathians exhibited increased velocities between 2007 and 2014, attributed to rising permafrost temperatures. Annual rates of horizontal surface kinematics of rock glaciers range from a few millimetres to a few meters (Strozzi et al., 2020), though

destabilization can result in velocities exceeding ten meters per year (Roer et al., 2008; Delaloye et al., 2013; Eriksen et al., 2018; Marcer et al., 2021; Hartl et al., 2023).

Many studies have demonstrated the feasibility of satellite radar interferometry (InSAR) for kinematic analysis of rock glaciers, capable of detecting motion at the millimetre scale (Liu et al., 2013; Necsoiu et al., 2016; Strozzi et al., 2020; Bertone et al., 2022). This technique enables the mapping of land surface deformation with an appropriate spatial and temporal resolution over vast areas (Bertone et al., 2022). Surface displacements can be attributed to permafrost creep only if the flow direction and velocity remain spatially consistent and uniform over a documented period (RGIK, 2023a). Permafrost creep typically occurs when the thickness of the ice-rich core in rock glaciers reaches at least 10-25 m (Cicoira et al., 2021). In contrast, displacements observed in rock glaciers with thinner layers of frozen debris are primarily driven by deformations within the active layer above the permafrost table.

While rock glaciers in discontinuous permafrost have been extensively studied, the distinctive behaviour of those in marginal periglacial environments has received far less attention (Serrano et al., 2010; Necsoiu et al., 2016). In such settings, rock glaciers exhibit either no movement or significantly slower movement rates (a few cm yr$^{-1}$) and are often referred to as transitional rock glaciers (RGIK, 2023a). This reduced surface velocity is attributed to the high shear strength of the material, which inhibits fast creep movement (Cicoira et al., 2021). Even if slow-moving rock glaciers were documented in various regions of the world (Brencher et al., 2021; Bertone et al., 2022; Lilleøren et al., 2022; Lambiel et al., 2023), the relationship between their velocity and ground ice content was rarely addressed (Serrano et al., 2010). Since borehole information is limited in high and remote mountains, a promising alternative to quantitatively estimate ground ice content is using petrophysical joint inversion (PJI) of seismic refraction and electrical resistivity data (Wagner et al., 2019).

The Southern Carpathian range is a key region in Europe where transitional rock glaciers are studied. Here, the enhanced continentality effects induce a distinct pattern of manifestation of periglacial phenomena compared with other mid-latitude mountains in Europe (Onaca et al., 2017a). In marginal periglacial mountains, permafrost occurrence is generally sporadic or patchy and site-specific characteristics highly control its distribution (Stiegler et al., 2014; Onaca et al., 2015; Kellerer-Pirklbauer, 2019; Popescu et al., 2024). Above 2000 m in the Southern Carpathians, the 1991-2020 climatological period was 0.8 °C warmer than the 1961-1990 baseline (Berzescu et al., 2025).

The paper aims to present new results on the rock glaciers dynamics and permafrost characteristics in the Retezat Mountains and, more broadly, to better understand the behaviour of rock glaciers in marginal periglacial mountains. To achieve this goal, we will (i) locate and assess the kinematics of rock glaciers` moving areas using SAR-based persistent scatterers interferometry (PSInSAR); (ii) update the existing rock glacier inventory in the Retezat Mountains with information on the rock glacier dynamics; (iii) estimate ground ice content using petrophysical joint inversion based on electrical resistivity and seismic refraction data and (iv) characterise the thermal conditions at the rock glaciers surface.

## 2. Study area

The Retezat Mountains are among the highest massifs in the Romanian Carpathians, constituting a distinct part of the Southern Carpathians (the latter are also known as the Transylvanian Alps). Located in the western part of the Southern Carpathians (45°22' N and 22°53' E), the Retezat Mountains reach an elevation of 2500 m, revealing a typical alpine landscape (Fig. 1). The climate in this region can be characterised as a moderate temperate continental climate, classified within the subarctic or boreal category according to the Köppen classification system. Between 2000 and 2100 m elevation, the mean annual air temperature is approximately 0° C, annual precipitation averaging around 1000 mm (calculated for the period 1961-2007) (Onaca et al., 2017a).

The mountain range is part of the orogenic units spanning two distinctive tectonic-structural regions: the Danubian Domain and the Getic Domain, both with the status of a thrust sheet. The Danubian Domain, referred to as the Danubian Autochtonous, is primarily characterised by two dominant granitic bodies, Retezat and Buta (Pavelescu, 1953). These granitic bodies are surrounded in the marginal area by epi- and meso-metamorphic schists, typifying the Getic Nappe. Specific Mesozoic rocks, particularly limestones, are prevalent in the southern part of this mountain range (Urdea, 2000). 87 % of the rock glaciers in the Retezat Mountains have developed on granite bedrock (Fig. 2), while the remaining landforms are situated on metamorphic schists.

The Retezat Mountains boast one of the most comprehensive and distinct arrays of glacial and periglacial landforms in the Romanian Carpathians. Notably, they host the largest glacial cirques in the Romanian Carpathians, with a combined area of

113  all glacial cirques amounting to c. 8 % of the massif's total area (Urdea, 2000). During the Last Glacial Maximum (LGM) (20.6

114  ka), glaciers in these mountains reached lower elevations ranging from 1000 to 1300 m (Ruszkiczay-Rüdiger et al., 2021).

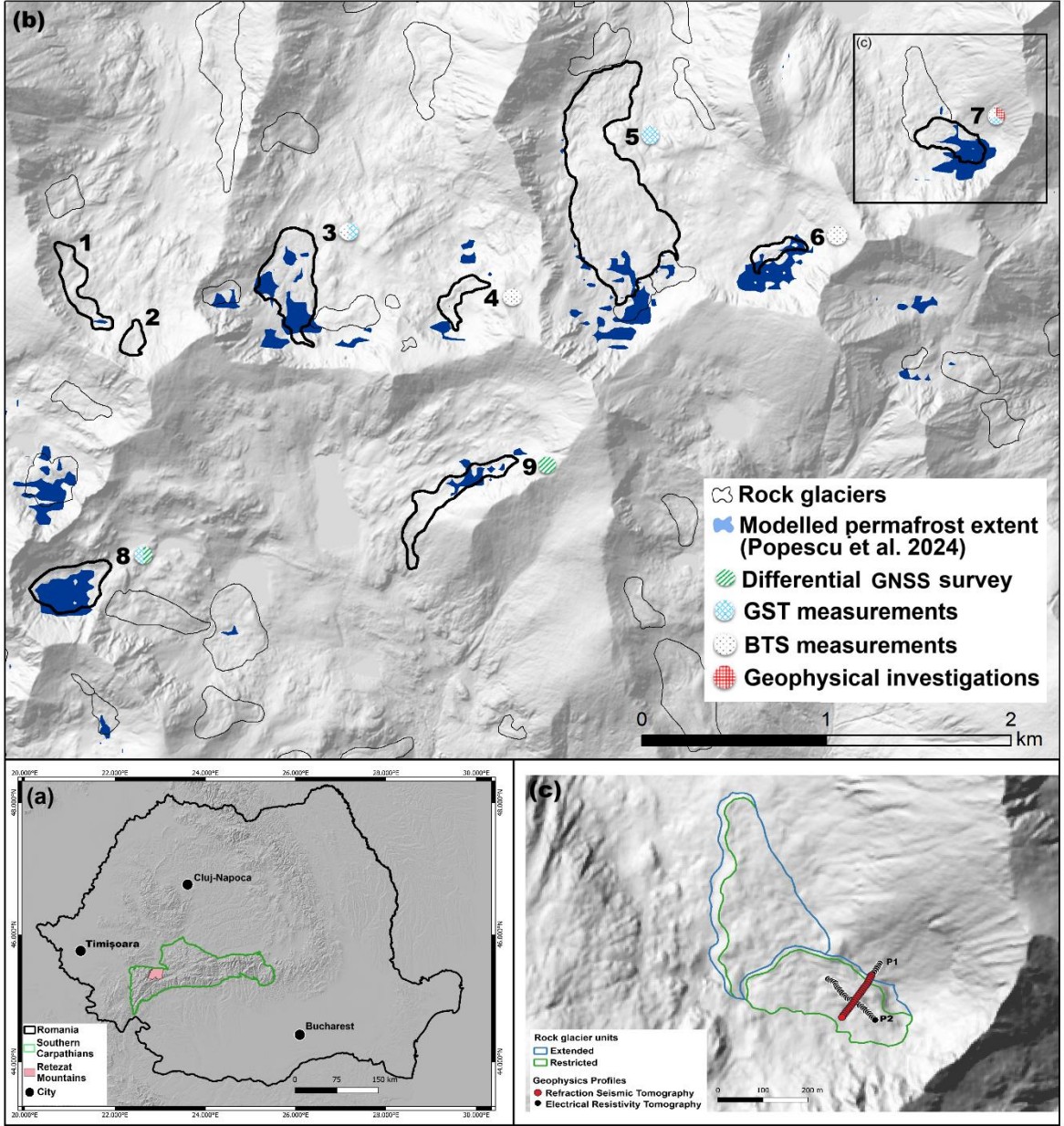

115

**Figure 1: Study sites. (a) Overview map with the location of the Retezat Mountains in the Southern Carpathians and in Romania, background of the map: hillshade based on FABDEM (Howker et al. 2022). (b) modelled permafrost extent (Popescu et al., 2024) and spatial distribution of rock glaciers in the Retezat Mountains overlaid on a hillshade based on the LAKI II DEM (LAKI II MNT, 2024). The rock glaciers that are discussed in the present paper are numbered (1 - Stânișoara, 2 - Bucura, 3 - Pietrele, 4 - Pietricelele, 5 - Valea Rea, 6 - Păpușa, 7 - Galeșu, 8 - Judele, 9 - Berbecilor), and the ground based measurements that have been performed on each of them are represented by a composite symbol near the number. (c) A detailed map with the position of the geophysics profiles on Galeșu rock glacier; note: same background image as (b).**

Subsequently, the Late Glacial period witnessed five phases of deglaciation. However, no glacial advance has been
documented in the central part of the Retezat Mountains during or after the Younger Dryas based on cosmic-ray exposure
dating (Ruszkiczay-Rüdiger et al., 2021). Rock glaciers in the Retezat Mountains likely began to develop during the Younger
Dryas, with most having become relict or transitional since the onset of the Holocene. Permafrost associated with rock glaciers
had been documented since 1993 in this mountain range (Urdea, 1993). A recent inventory described Retezat Mountains as
the range with the highest number (94) and density (0.52 landforms/km$^2$, and 2.87 ha/km$^2$ at altitudes above 1540 meters) of
rock glaciers in the Romanian Carpathians (Onaca et al., 2017b) (Fig. 1). Additionally, they harbour the longest Carpathian
rock glacier, Valea Rea, extending 1.4 km (Urdea, 2000) (Fig. 2b).

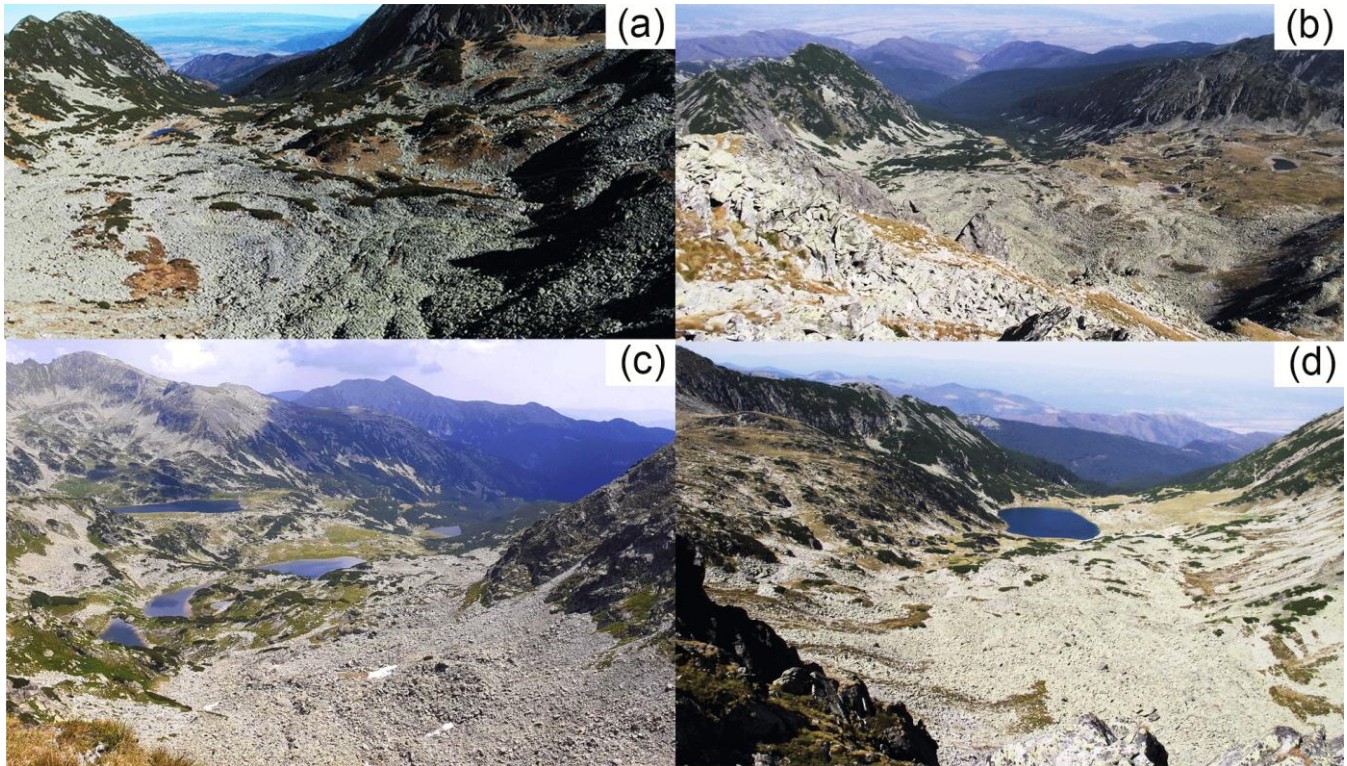


**Figure 2: Pictures of the rock glaciers in the Retezat Mountains: (a) Pietrele; (b) Valea Rea; (c) Judele; (d) Galeşu. Photo credit: A.**
**Onaca.**
**3. Methods**
**3.1. Rock glacier inventory**
Rock glaciers are categorised into three types based on their activity status: active, transitional, or relict, as outlined by (RGIK,
2023a). Active rock glaciers exhibit consistent downslope movement across most of their surface with displacement rates
ranging from a decimetre to several meters per year (RGIK, 2023a). Most of the surface of a transitional rock glacier
experiences little to no downslope movement, with annual average displacement rates generally falling below one decimetre

(RGIK, 2023a). Rock glaciers exhibiting no detectable movement across most of their surface are classified as relict (RGIK, 2023a).

This study revised the existing inventory of rock glaciers in the Southern Carpathians (Onaca et al., 2017b) in accordance with the guidelines provided by RGIK (2023a). The inventory involved mapping rock glaciers through fieldwork surveys and detailed examination of high-resolution aerial imagery. Due to the availability of kinematic information for only a limited number of landforms (Vespremeanu-Stroe et al., 2012; Necsoiu et al., 2016), the current inventory lacks data on the activity of rock glaciers. Information on rock glacier kinematics was only available for a few landforms (Vespremeanu-Stroe et al., 2012; Necsoiu et al., 2016), while most of the rock glaciers were classified as either intact or relict based on geomorphological and ecological criteria (e.g., degree and type of vegetation cover).

### 3.2. Persistent scatterer interferometry using Sentinel-1 data

PSInSAR is a remote sensing technique designed to measure ground displacements along the radar line of sight (SAR LOS) with millimetric accuracies (Rucci et al., 2012; Yu et al., 2020). Although Sentinel-1 (S1) SAR data does not offer the highest possible spatial resolution, its worldwide periodic coverage and open data policy has enabled wide-scale monitoring since 2014, leading to a thriving archive of ground-motion products with various applications.

Sentinel-1 serves as the backbone of the operational PSInSAR application development for the European Ground Motion Service (EGMS), openly available throughout the entire European area. The PSInSAR analysis employed in this study was developed by Terrasigna and generally follows the EGMS specifications (https://land.copernicus.eu/en/technical-library/egms-algorithm-theoretical-basis-document/@@download/file). However, there are a few differences, particularly related to the choice of reference points, the modelling of atmospheric effects in steep terrain and the selection of SAR images. EGMS products are computed at the regional level, where reference points are typically located in lowland areas covered by infrastructure, which provides strong and stable radar backscattering. Additionally, EGMS includes all available acquisitions, even those affected by snow cover at high altitudes. However, inspection of EGMS products reveals that extending the measurement network from lowland reference points to mountain summits was largely unsuccessful. This is mainly because the atmospheric path delay associated with steep topography was not adequately compensated for and acts like phase noise. Furthermore, snow cover during winter significantly impacts data quality. Non-homogeneous snow or snow with variable humidity scatters radar signals and increases phase noise. If the case of dry snow, radar waves penetrate the snowpack, but because they propagate more slowly than in air, the interferometric phase experiences a time delay. This delay produces apparent subsidence (false ground displacement away from the radar sensor). Combined these factors often result in the rejection of radar targets on mountain tops due to excessive noise.

To address these issues, Terrasigna carefully selected reference points located on mountain summits, where the topographic atmospheric delay is similar to that of the areas of interest. Additional efforts were made to improve the accuracy of atmospheric delay modelling and compensation. Furthermore, only snow-free acquisitions were considered. As a result, the high density of radar targets – formed by bare rocks at the top of the mountains – is preserved in our measurements.

Both ascending and descending paths were processed for cross-validation, along with L-band ALOS data, which was analyzed for the same purpose. Because descending passes occur in the early morning – when atmospheric conditions are generally more stable than in the evening – the resulting measurements tend to be less noisy. A 2D decomposition between ascending and descending passes is technically feasible; however, the steep topography introduces several challenges. First, areas that are clearly visible from one orbit may be in shadow or appear foreshortened in the other, reducing data quality and spatial consistency. Second, since the topography is steep, the preferential direction of ground movement is often dictated by the slope of the terrain. Additionally, the 2D decomposition estimates vertical and east-west displacement components under the assumption that there is no north-south movement – an assumption that is frequently invalid in mountainous regions, where north-south displacement is commonly observed. Based on these issues, it was decided to use the orbits that yielded the best results for validation and mapping.

Figure 3a illustrates the total coverage (from all available S1 paths from both the Ascending and Descending orbits) of the EGMS product in the area of interest, while Figure 3b illustrates the PSInSAR analysis of the same area, derived solely from one S1 path (Path 80 Descending). In the following, this is referred to as `Terrasigna PSInSAR`. It is evident that the EGMS coverage is relatively sparse and does not highlight dynamic areas – there are no zones marked in red, which typically indicate significant ground motion. In contrast, the PSInSAR results from Terrasigna show much denser coverage and clearly identify dynamic areas, with red colors representing higher displacement rates. In general, there are technical differences in the computation of data across EGMS products, primarily due to the involvement of multiple groups in the project. Terrasigna's algorithms are more closely aligned with those used for the EGMS in southern Europe, which appear to offer better extraction of non-linear motion.

In this study, the kinematics of the rock glaciers were assessed using 181 images acquired between May 15, 2015, and October 4, 2021, covering only the snow-free periods to avoid coherence loss. The motion was measured along the SAR LOS direction; however, the actual displacement of the rock glacier surface is expected to mainly occur along the slope or in the vertical direction. The PSInSAR algorithm, as described by Rucci et al. (2012), Crosetto et al. (2016) and Poncoș et al. (2022), preserved all displacement information to maximize the chances of detecting slow movements (mm yr$^{-1}$) in areas without vegetation cover. The process began by extracting linear deformation information before applying any spatial or temporal filtering, which was typically used to improve phase statistics. A key challenge is that the atmospheric phase is two orders of magnitude larger than the displacement signal (Poncoș et al., 2022), requiring meticulous phase unwrapping and correction of each residual interferogram. Due to significant atmospheric noise in steep terrain or in areas with large elevation differences, a reference point at a similar elevation to the rock glaciers was chosen on the mountain summit. This approach minimizes atmospheric phase differences, improving coherence and reliability of the PSInSAR measurements. An important number of rock glaciers in the area are oriented north-south or south-north, which may lead to an underestimation of actual displacements due to the limited sensitivity of the satellite look angle to slope-parallel motion.

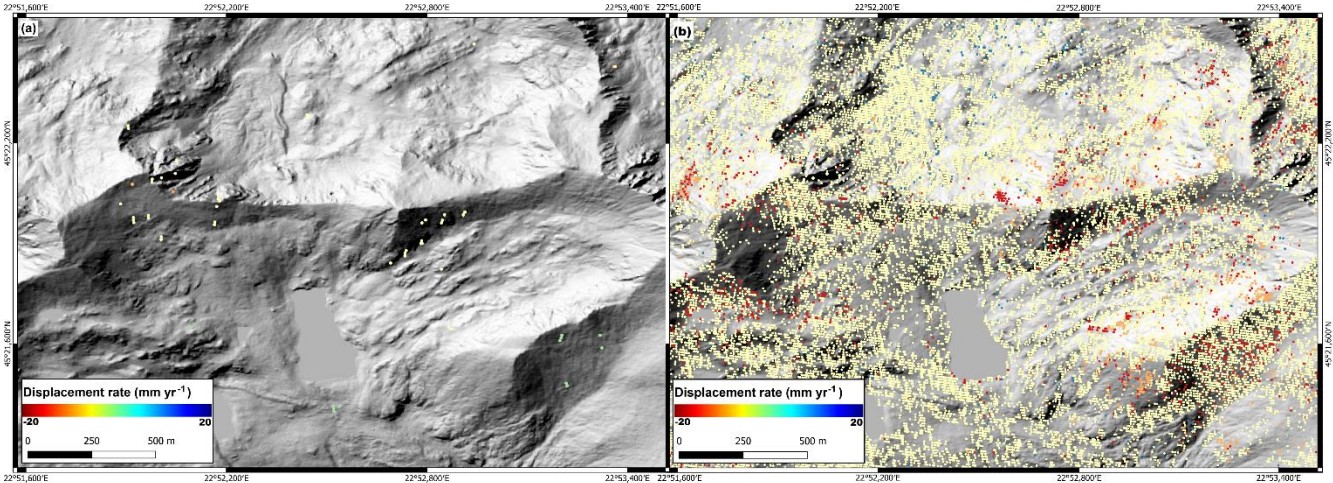


**Figure 3: Comparison between the PSInSAR spatial density of measurements obtained by EGMS (a) and Terrasigna (b) in the**
**central area of Retezat Mountains. Background of both maps: hillshade based on the LAKI II DEM (LAKI II MNT, 2024).**
The PSInSAR results were analysed using the Persistent Scatterers Online Software Tool (PSTool), a web-based platform
developed by Terrassigna Inc. (Poncoș et al., 2022), to exploit a large volume of ground displacement data. The PSTool
platform can be used to inspect temporal characteristics of the ground motion, select areas of interest and extract temporal
averages of displacement rates, export temporal profiles to standard formats for integration in the user's own platforms and
upload user-specific layers on top of the displacement information.
**3.3. Inventorying moving areas**
According to RGIK (2023a) guidelines, a moving area (MA) represents an area at the surface of the rock glacier characterised
by relatively homogeneous velocity rates and consistent flow direction. Based on multi-annual surface velocity rates, MAs
were identified and delineated within the inventoried rock glaciers (Onaca et al., 2017b) using Terrasigna PSInSAR results.
The next step was to assign velocity classes to moving areas considering the standardised velocity classes (Barboux et al.,
2014; Bertone et al., 2022). In our case, MAs were attributed to one of the following SAR LOS deformation velocity classes:
undefined, 0.3-1 cm yr$^{-1}$, 1-3 cm yr$^{-1}$, and 3-10 cm yr$^{-1}$ (RGIK, 2023a) (Fig. 4). The velocity class
characterizes the average yearly displacement rate recorded within a MA during the 2015-
2021 period. The undefined category was assigned to MAs characterised by inhomogeneous
velocity rates. PSInSAR-based surface displacement of ≤ 0.3 cm yr$^{-1}$ were assigned to the "no
movement" category, as this threshold was considered the lower limit of velocity detection on S1 interferograms in this type
of approach (Rouyet et al., 2021). The moving areas were manually digitized and compiled into an inventory using ArcGIS
10.8. Since many surface displacements in marginal periglacial regions result from active layer deformations (e.g., melt-induce
subsidence), we have set the minimum area for an MA at 1000 m$^2$ to exclude those not associated with permafrost creep. For
the spatial analysis of the rock glaciers and MAs, we used a one-meter resolution digital elevation model generated from high-
resolution LiDAR source data acquired in 2018 (LAKI II MNT, 2024).

### 3.4. Differential GNSS measurements

Judele (8) and Berbecilor (9) rock glaciers had been surveyed by differential GNSS (DGNSS) measurements every summer
between 2019 and 2021. A differential dual-frequency Topcon Hiper V GPS had acquired high-precision positioning data in
real-time kinematics mode. The dGNSS device uses two receivers, one installed as a fixed base station, whereas the roving
receiver was moved to different points of interest in the field. The mobile receiver gets the corrected position information
calculated by the base station via a radio signal in order to measure a point with very high precision (i.e. < 1 cm accuracy in
the horizontal plane). 25 survey markers were measured in October 2019 and remeasured in October 2020 and 2021. Two
points outside the boundaries of the rock glaciers, located on stable bedrock, were also measured to assess the horizontal
accuracy of the DGNSS. The difference between the yearly measurements in both cases indicated an accuracy range of 0.3 to
0.6 mm/yr$^{-1}$. The mean DGNSS velocities used in the analysis were calculated as the yearly displacement between the initial
and final position over a two-year period.

### 3.5. Validation with ALOS-2 PALSAR-2 interferometry

To further validate Terrasigna's PSInSAR analysis specifically developed for this study, we considered a series of six ALOS-
2 PALSAR-2 images regularly acquired between 2014 and 2019 at the end of the snow-free season in September and October.
We computed wrapped differential interferograms with time intervals ranging from one to five years using a DEM with 10 m
pixel spacing obtained from 1:25 000 scale topographic maps with a contour interval of 10 m. For the interpretation of the
interferograms, we followed the practical guidelines of the IPA Action Group Rock glacier inventories and kinematics (RGIK,
2023b).

### 3.6. Thermal conditions

The bottom temperature of the winter snow cover (BTS) is an efficient method to map permafrost distribution in non-arid
mountains (Vonder Mühll et al., 2002). If optimum snow conditions are met, the BTS values indicate probable permafrost at
< -3 ℃, possible permafrost at -2 to -3 ℃ and absence of permafrost at > -2 ℃ (Haeberli, 1973; Hoelzle, 1992; Popescu et al.,
2024). However, in dry, porous bouldery surfaces where air convection and advection occur, this method lacks the precision
needed to accurately map permafrost occurrence (Bernhard et al., 1998). Nevertheless, the BTS method remains highly
effective for distinguishing areas with colder ground surface temperatures from those with warmer ones. Two classical 2.6 m
long BTS probes equipped with digital thermometers (0.5 ℃ precision) were used to measure 140 thermal records at the snow-
ground interface. The BTS measurements were acquired at the end of March 2022 on four rock glaciers in three north-facing
valleys in the central part of the Retezat Mountains (Fig. 1). At all the sites where BTS values were recorded, the snow was
sufficiently thick (> 80 cm) to insulate the ground from external air temperature fluctuations (Ishikawa et al., 2003). Previous

studies in the Southern Carpathians (Vespremeanu-Stroe et al., 2012; Onaca et al., 2015) revealed that in March, BTS values remain nearly constant below a thick snow cover, which usually falls in November or December.

Minimal temperature data loggers became widely used in mountain permafrost terrain to get more detailed insights into the energy exchange fluxes at the surface of rock glaciers (Hoelzle et al., 1999). Four rock glaciers in the central part of the Retezat Mountains were selected to monitor the thermal regime at the ground surface (Fig. 1b). The evolution of ground surface temperature (GST) was recorded using iButtons DS1922L data loggers. According to the producer, the miniature thermistors used in this study have an accuracy of $\pm$ 0.5 ºC and measure temperatures between -40 and 80ºC. The sensors were indirectly calibrated at 0 ºC using the snow melting period ("zero curtain" interval), and GST data were measured and logged every 2 hours. In mid and late winter, the ground surface temperature remains stable under a thick insulating layer, and the subsurface mainly controls the energy flux. This is why the "winter equilibrium temperature" (WEqT) is considered an excellent empirical predictor of permafrost existence if temperatures are low (i.e., < -2 ºC) (Sattler et al., 2016). WEqT refers to stable ground surface temperature period lasting at least two weeks, generally occurring in late winter, when snow cover exceeds 50 cm in thickness (Schoeneich, 2011). WEqT and mean annual ground surface temperature (MAGST) were calculated for each GST monitoring site.

### 3.7. Geophysical Methods and PJI Modelling

Geophysical methods, such as electrical resistivity tomography (ERT) and refraction seismic tomography (RST), are widely applied in mountain permafrost studies and have the potential to characterise subsurface structure and heterogeneity and detect and map ground ice occurrences (Hauck et al., 2011; Herring et al., 2023). Both methods are sensitive to differences between frozen and unfrozen subsurface conditions. As ice can be considered an electrical insulator as opposed to water, the electrical resistivity increases exponentially with decreasing temperatures below 0 °C. Similarly, the seismic P-wave velocity of ice is with 3500 m s$^{-1}$ significantly higher than that of liquid water ($\sim$ 1500 m s$^{-1}$) or air (330 m s$^{-1}$), allowing to differentiate between frozen sediments (containing ice) and unfrozen sediments (pore space filled with water or air).

ERT is the most common geophysical technique applied in permafrost research and is used for mapping permafrost occurrence where no borehole information is available, as well as monitoring changes in the ice-to-water ratio (Wagner et al., 2019; Mollaret et al., 2020). The RST method is often used as a complementary method to ERT to reduce the ambiguity in the interpretation of ERT data, as the P-wave velocity $v_p$ is mainly controlled by density, and variations in $v_p$ allow to identify porosity changes, or discriminate between liquid (water) and solid (ice) pore fluid, as well as in determining the depth to the bedrock.

In the absence of ground truth information about the state of permafrost, another advantage of geophysical data is, that co-located ERT and RST data can be used to quantitatively estimate the content of the four phases (i.e., rock, ice, water and air) using the so-called 4-phase model approach, which is based on the petrophysical equations by Archie (1942) for the electrical resistivity and that of Timur (1968) for the P-wave velocity.

Recently, the approach has been further developed by Wagner et al. (2019), to the so-called petrophysical joint inversion (PJI) framework, permitting the joint inversion of ERT and RST data sets to simultaneously solve for the subsurface distribution of the 4 phases. The main advantage of the PJI is the increased accuracy of the parameters estimated, as the algorithms iteratively search for a subsurface model that simultaneously explains the seismic and resistivity measurements. This is especially relevant for the porosity model, which is represented more realistically in the PJI than in previous versions of the 4-phase model (Hauck et al., 2011). Mollaret et al. (2020) demonstrated the applicability of the PJI for data collected on different alpine permafrost landforms with different ice contents.

In the field, 2D ERT data were collected using a GEOTOM (Geolog) multi-electrode instrument equipped with 50 electrodes spaced 4 meters apart. By combining a multitude of individual measurements with different electrode combinations (i.e. quadrupoles) along a profile line, a 2-dimensional resistivity model of the subsurface is obtained. All measurements were performed in the Wenner configuration to ensure an optimal signal-to-noise ratio, which is especially important in dry and coarse-blocky terrain.

2-dimensional RST data were obtained using a 24-channel Geode seismometer (Geometrics). An artificial seismic wave is produced by hitting a sledgehammer to the ground, and the waves travel along different paths through the subsurface and back to the surface, where they are registered by 24 geophones. The subsurface structure and composition can be derived from the travel time the so-called P-wave needs from the source (i.e. hammer) to the geophones. The wave velocity ($v_p$ in m s$^{-1}$), and thus the travel time, is basically a function of the density of the subsurface material, and the obtained seismic velocity allows inferring the subsurface material. The pre-processing of the seismic field data (picking of first arrivals) was performed using the software *ReflexW* (Sandmeier, 2020). The individual ERT and RST data sets were first independently inverted using the PyGimLi framework (Rücker et al., 2017), and in a second step, the PJI was conducted to estimate ground ice contents using the approach developed by (Wagner et al., 2019).

## 4. Results

### 4.1. Inventorying moving areas

In the Retezat Mountains the MAs exhibited velocities ranging from 0.3 to 5 cm yr$^{-1}$ (Fig. 4 and 5), and were classified into three velocity classes: 0.3 – 1 cm yr$^{-1}$, 1 – 3 cm yr$^{-1}$, and 3 – 10 cm yr$^{-1}$. The measured displacement between 2015 and 2021 remained consistent for most MAs, with no significant changes in velocity during this period (see trendlines in Fig. 5). Due to the relatively low velocities of the MAs, tracking seasonal variations involved high uncertainty; therefore, we referred only to annual or multiannual displacement rates.

A total of 92 MAs covering 0.27 km$^2$ were inventoried in the Retezat Mountains rock glaciers. Most of the MAs are classified with the slow velocity class (0.3 – 1 cm yr$^{-1}$ and 1-3 cm yr$^{-1}$) (Fig. 6), while only 10 % of the MAs are characterised by velocity class 3-10 cm yr$^{-1}$ (Fig. 6a). Around one-third (37 %) of the total number of rock glaciers in the Retezat Mountains contains MAs, but the analysis revealed they usually occupy only a small portion (< 30 %) of the total surface of each rock glacier; in

six cases, the cumulated area of MAs represents more than 50 % of the rock glacier area (Fig. 6d). The mean area of MAs is
0.3 ha, ranging from 0.1 to 1.77 ha.
The number of MAs in each rock glacier varies between 1 and 8, but in most cases (69 %), 1 to 3 MAs occur in an individual
RG. MAs characterised by velocities > 3 cm yr$^{-1}$ were identified in 8 rock glaciers, whereas MAs classified in the velocity
class 1-3 cm yr$^{-1}$ appear in 17 (Fig. 6c).
The median elevation for each MA class falls between 1950 and 2295 m (Fig. 7). Specifically, 62 % of the MAs are found in
the elevation band of 2100 to 2200 m, while 17 % lie between 2200 and 2295 m. Additionally, 16 % of the moving areas are
situated in the range of 2000 to 2100 m, and only 5 % are below 2000 m. Among these, MAs categorised under velocity classes
of 1-3 and 3-10 cm yr$^{-1}$ generally occur at the highest elevations (Fig. 7a). Figure 7b illustrates the variability of slopes across
the MAs velocity classes, revealing mean values ranging from 8 to 42°. The widest range of slopes is observed in the velocity
class 0.3 – 1 cm yr$^{-1}$, which also exhibits higher median values. Half of the MAs (50 %) face north (Fig. 8), despite that only
21 % of the inventoried rock glaciers in the Retezat Mountains stand out on the northern aspects. The NE and E slopes host
more MAs (32 %) compared to NW and W aspects (23 %) in respect with the rock glacier distribution (Fig. 8). Across the
mountain range, slopes with a western aspect dominate in terms of surface coverage. The MAs with velocities exceeding 3 cm
yr$^{-1}$ receive the lowest potential solar radiation (Fig. 7c).

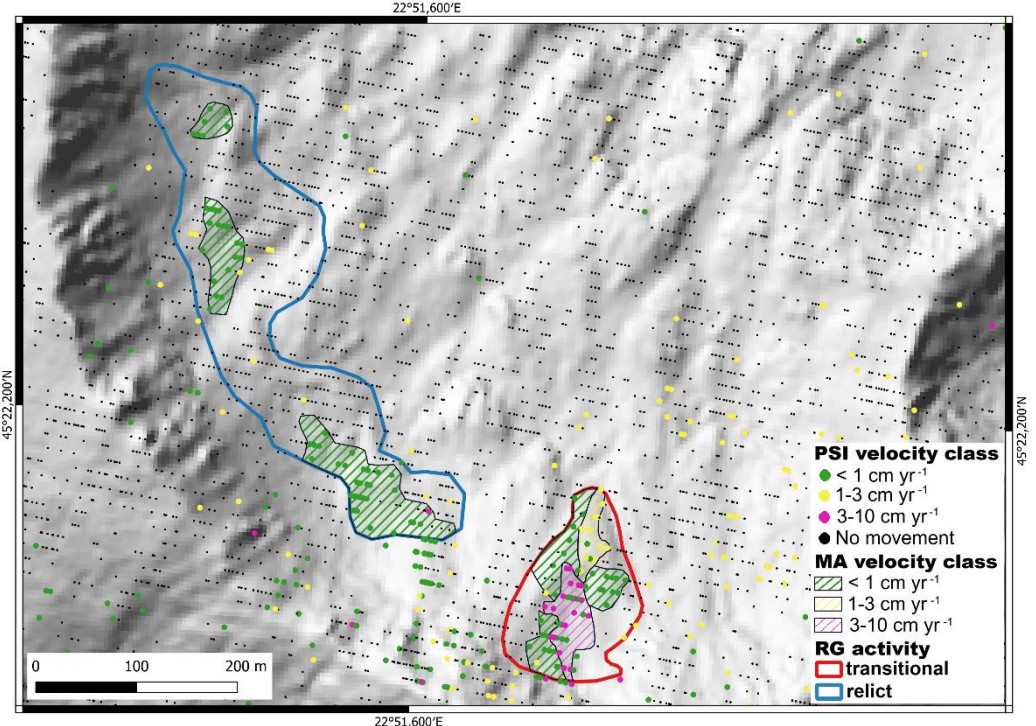


**Figure 4: Example of moving areas and rock glacier activity for the Retezat Mountains. The two RGs are marked with numbers 1**
**and 2 in fig.1. Background image: hillshade based on the LAKI II DEM (LAKI II MNT, 2024).**

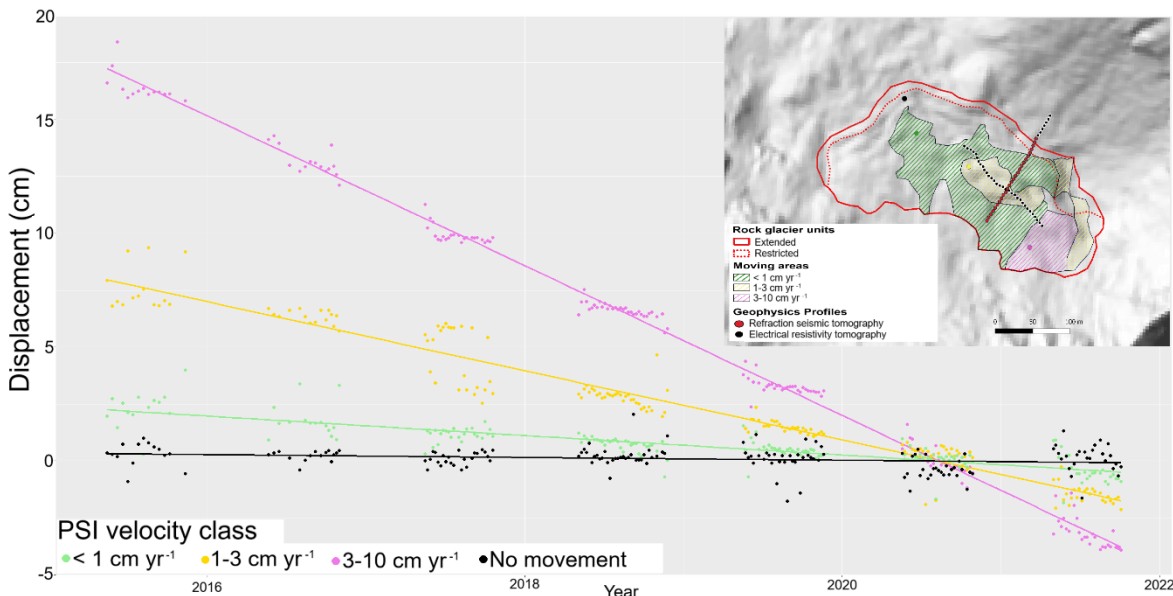


**Figure 5: Displacement profiles over a period of 6 years (2015 – 2021) for 4 locations (identified in the location map with dots of corresponding colour) representing each velocity class and one for an area with no movement. The dots show the actual PSI displacement results, while trend lines (linear regressions) indicate long-term motion patterns. The displacement is measured relative to 2021. The downward trend can be interpreted as movement away from the sensor, which in our case represents a combination of vertical movement and horizontal downslope movement. The gap in point density along the trend line is due to the winter season, the measurements being performed in snow free conditions, usually from June to November.**

For rock glaciers exhibiting no or minimal movement (< 1 cm yr[-1]), the RGIK (2023a), recommends assigning a relict activity class. The present analysis shows that only 21 % of the rock glaciers in the Retezat Mountains could be classified as transitional (Fig. 9a), encompassing areas with moving velocities ranging between 1 and 5 cm yr[-1]. The transitional rock glaciers exhibit a median elevation of 2170 m, surpassing that of the relict ones by 150 m (Fig. 9b). Additionally, the median size of transitional rock glaciers is slightly smaller than that of relict rock glaciers (Fig. 9c).

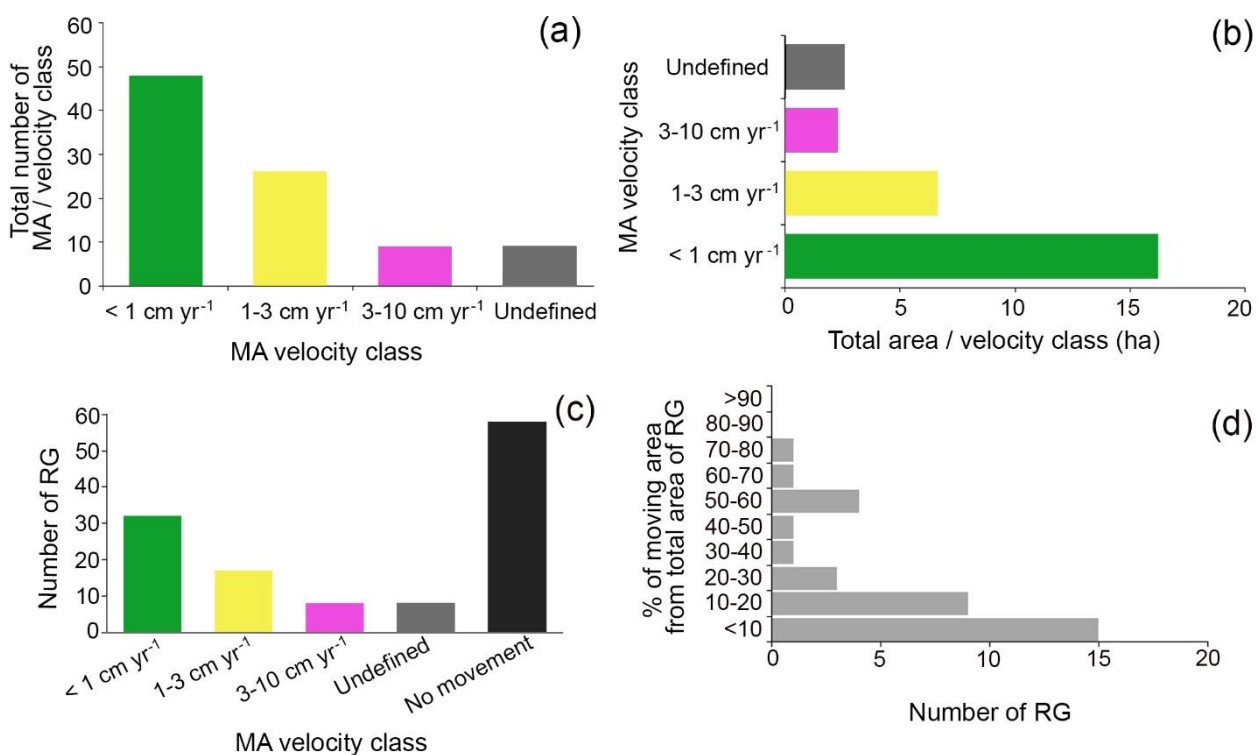


**Figure 6: The moving areas classified by velocity classes (a) and their extent (b). The number of rock glaciers containing moving**
**areas and without moving areas (c) and the percentage of the moving area cover within rock glaciers (d).**

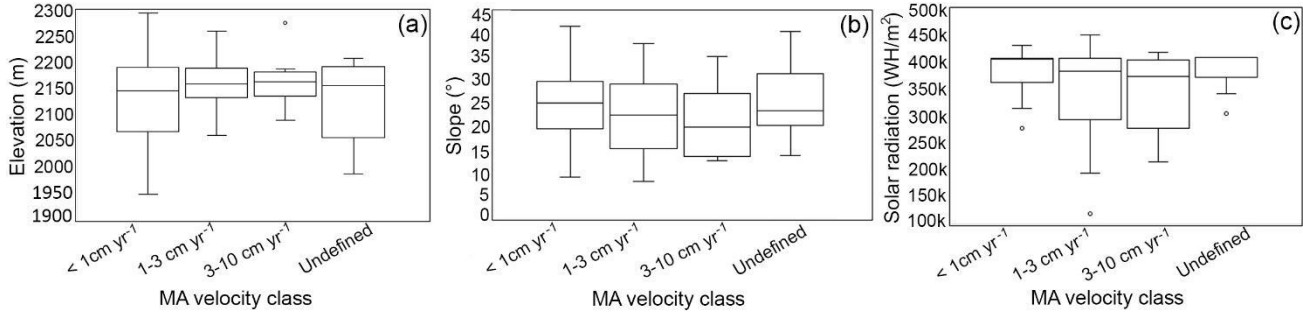


**Figure 7: Elevation (a), Slope (b) Potential solar radiation (c) vs MA velocity classes**

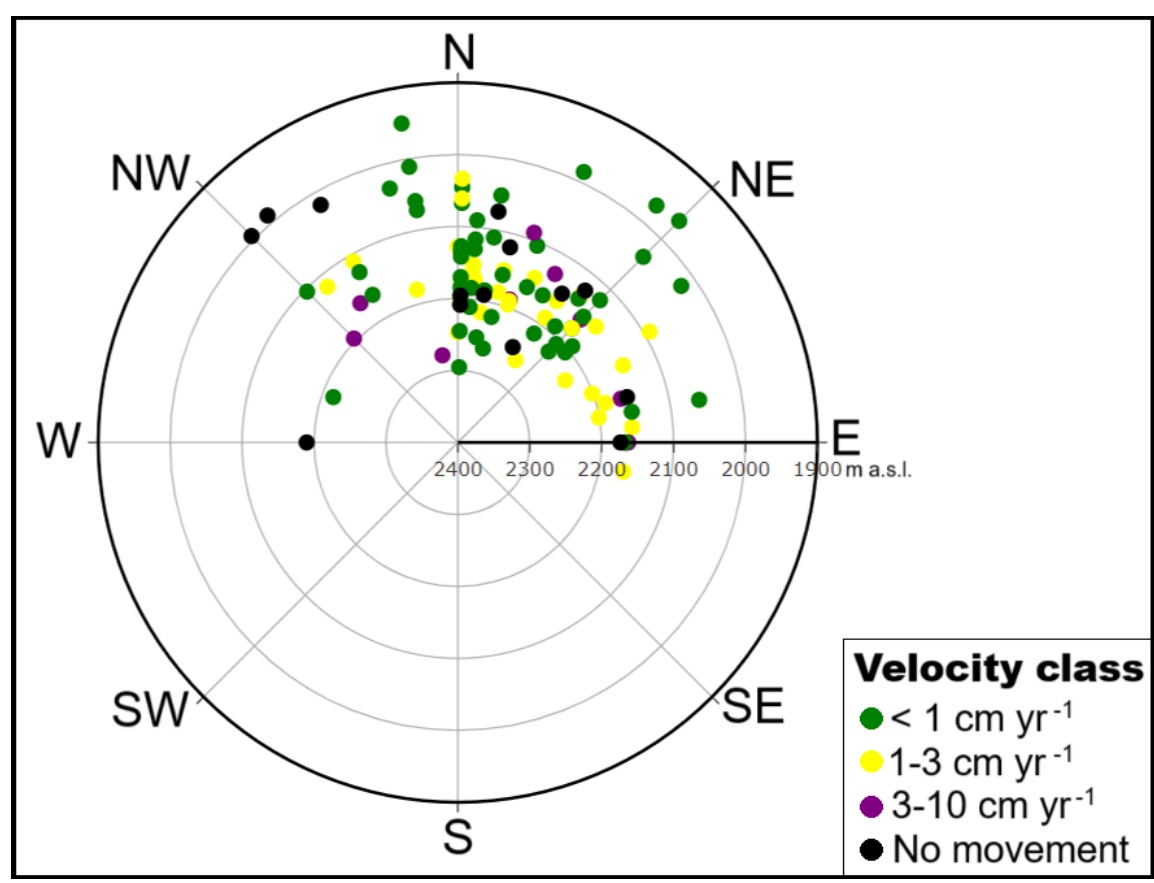


**Figure 8: Distribution of moving areas, in the Retezat Mountains, in relation to slope aspect (angular axis) and elevation (radial axis).**

Figure 10 presents a comparison between ALOS-2 PALSAR-2 interferogram and the Sentinel-1 PSInSAR results at Galeşu (7) site. Although the accuracy of the ALOS-2 PALSAR-2 is lower, both products exhibit similar signals. The main displacement areas are clearly visible and coincide on both maps

The comparison between PSI and InSAR performance reveals that, in areas with high variability in displacement, PSI provides more detailed mapping of the monitoring areas (MAs), allowing for the differentiation of relatively minor velocity differences. In Fig. 10b, a specific MA is clearly identified and classified as having a velocity of <1 cm yr$^{-1}$. The same area appears in Fig. 10a, where most of it is also mapped as <1 cm yr$^{-1}$; however, adjacent zones are classified as 1–3 cm yr$^{-1}$ and 3–10 cm yr$^{-1}$, indicating a broader range of detected velocities.

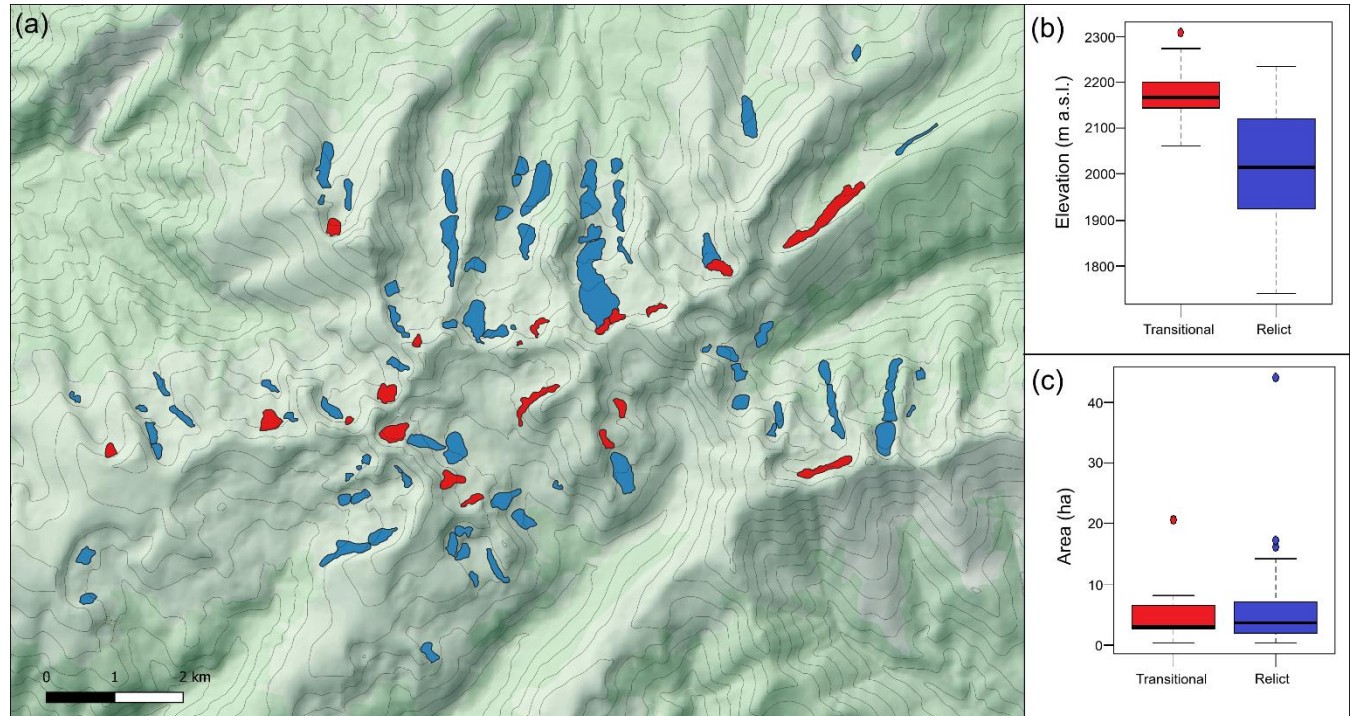

369

**Figure 9: The spatial distribution of transitional and relict rock glaciers in the Retezat Mountains (a) and their median elevation (b) and size (c).**

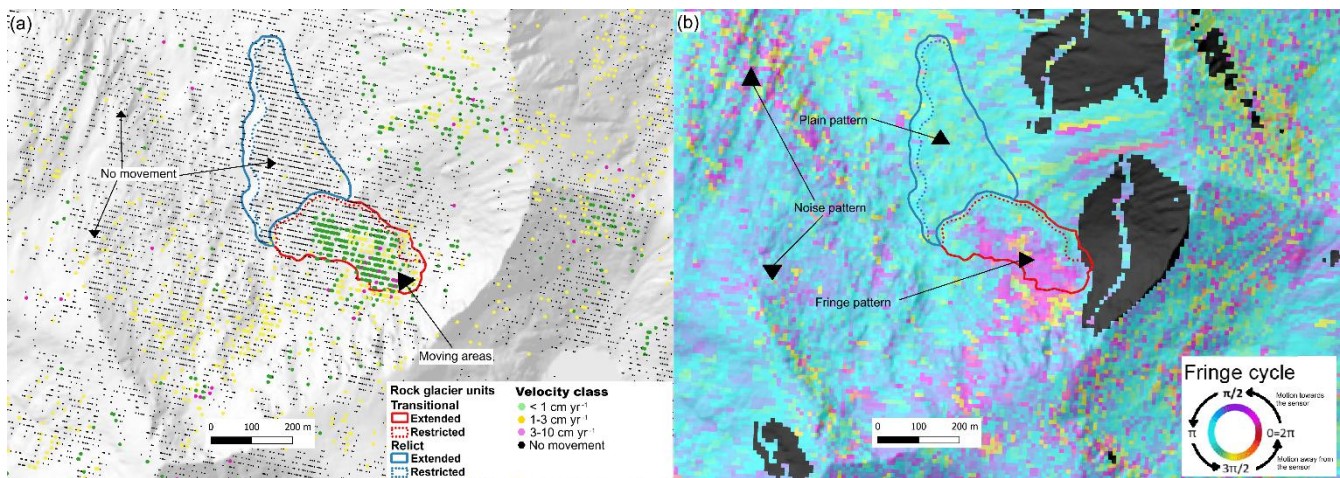

372

**Figure 10: A comparison between multiannual PSINSAR from Sentinel 1 (a) and InSAR from ALOS-2 PALSAR-2 (b). Notice the same moving areas inside the RGs, that are revealed by clustered pixels with movement (green, yellow and magenta) from PINSAR (a) and the areas with fringe patterns (b). In (b) the shadow areas are masked out and the fringe cycle (bottom right) represents the change of colour. Fig. 10b presents the Galeşu RG where in the upper RGU we have one fringe over a five year period (September 2014 to October 2019) accounting for a displacement of approximately 0.56 cm/yr. This is in line with the PSI measurements that have the biggest MA on the RG to be <1 cm/yr.**

### 4.2. GNSS measurements

The mean velocities measured by dGNSS ranged between 0.4 and 2.8 cm yr$^{-1}$, with values exceeding 1 cm yr$^{-1}$ for 56 % of the marker points (Fig. 11). On the Judele (8) rock glacier, eight marker points recorded velocities between 1 and 2.6 cm yr$^{-1}$, while nine points showed velocities between 0.4 and 1 cm yr$^{-1}$. The highest velocities are observed in the central part, gradually decreasing toward the margins. One marker point measured on the front of Judele rock glacier revealed very low rates of displacements (0.4 cm yr$^{-1}$). Almost all marker points indicate consistent movement toward the front of the rock glacier. Four marker points on the Berbecilor (9) rock glacier revealed velocities between 1 and 2.8 cm yr$^{-1}$ and three between 0.6 and 1 cm yr$^{-1}$. All the seven marker points indicate consistent movement toward the front. Most marker points with moving velocities between 1 and 2.8 cm yr$^{-1}$ (89 %) were located within MAs categorised under the 1-3 and 3-10 cm yr$^{-1}$ velocity classes by PSInSAR analysis. At Judele, five out of eight dGNSS markers recording velocities between 1 and 2.6 cm yr$^{-1}$, fall within the 1-3 cm yr$^{-1}$ velocity class, whereas at Berbecilor the ratio is two out of four. At both sites, nine marker points with velocities between 0.4 and 1 cm yr$^{-1}$ are located outside any designated MAs, while only three falls within the corresponding velocity class.

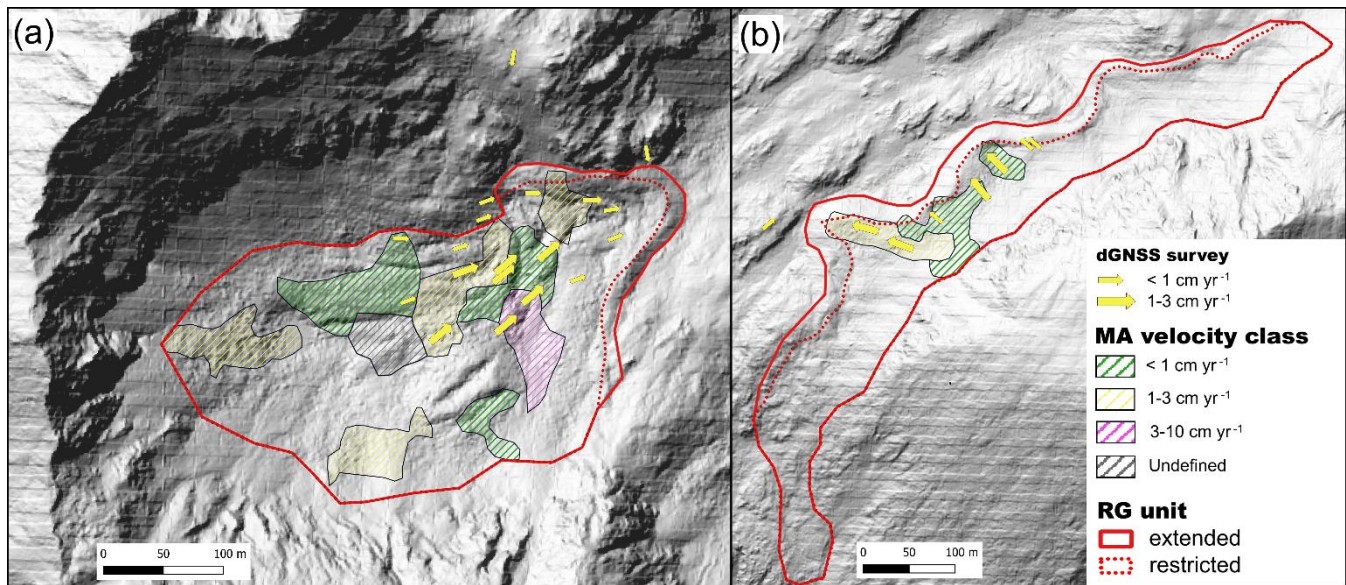

**Figure 11: Horizontal displacements derived from GNSS measurements for 2019-2021 at Judele (a) and Berbecilor (b) sites, overlayed on the MAs derived from PSInSAR. Background of both maps: hillshade based on the LAKI II DEM (LAKI II MNT, 2024)**

### 4.3. BTS and ground temperatures

BTS measurements allowed the identification of colder versus warmer ground surface areas in four rock glaciers (Fig. 12). Half of the 140 measured BTS points were measured on Pietrele (3) rock glacier, which has the lowest front altitude and revealed the warmest mean of BTS values (-2.9 °C). The coldest temperatures at this site occurred in the uppermost part, where

most BTS values were below -3 °C. Several BTS points measured on the talus slope feeding this rock glacier also revealed very low temperatures. In the western part, where a MA in the 0.3-1 cm yr⁻¹ velocity class is present, BTS values consistently fell below -5 °C. At the Galeșu multi-unit rock glacier, the mean BTS was -3.9 °C, with the coldest values clustered in the southern unit and on the upslope talus. The BTS values within the MAs were all very low, below -5 °C. In contrast, the northern rock glacier unit, which showed no signs of surface displacement, exhibited a highly heterogeneous distribution of BTS values. Păpușa (6) is the highest rock glacier where BTS measurements were performed and revealed the lowest average of BTS values (-4.1°C), whereas at Pietricelele (4), the mean BTS was -3.5 °C. At Păpușa site, a warmer zone occurred in the central part, with three BTS values warmer than -2 °C; however, all BTS values within the MAs indicated very cold ground conditions. Similarly, at Pietricelele, a warmer ground surface area was identified in the central and northeastern parts, while the western and southeastern areas revealed colder temperatures. Snow depth at the probing points ranged from 80 and 260 cm. In all the cases, the calculated median BTS in MAs was lower compared with the median of all BTS values in each site.

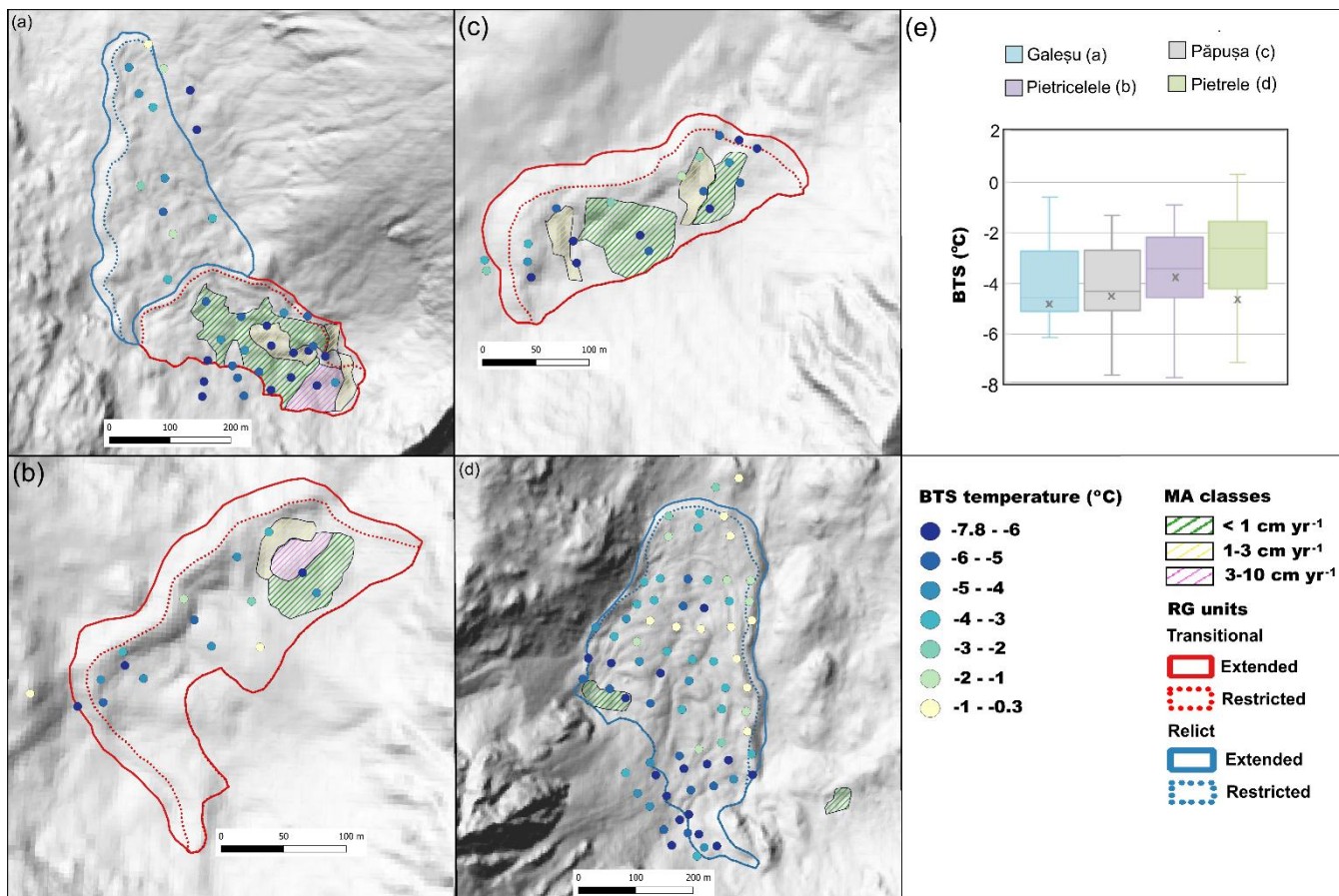

**Figure 12: BTS measurements performed in March 2022 on four rock glaciers in the Retezat Mountains: (a) Galeșu, (b) Pietricelele, (c) Păpușa, (d) Pietrele. (e) Summary box-plot diagram of the BTS measurements, the horizontal line drawn inside denotes the median BTS for each rock glacier, while the x represents the median BTS of the moving areas of each rock glacier). (f) the legend for the maps in (a), (b), (c) and (d).**

In most cases, MAGST values were negative at the monitoring sites from 2013 to 2022, ranging from -2.3 ℃ to 0.8 ℃, with the lowest values recorded at Galeșu and the highest at Pietrele. To illustrate long-term GST evolution, Figure 13a shows the running annual mean of ground surface temperature. However, subzero MAGST values were recorded only at site Valea Rea (5) in all the years, whereas at Galeșu, all the MAGST values were below 0 ℃ except one year (e.g., 2016). All the GST sites revealed negative values for the mean temperature of the entire monitoring interval.

Fig. 13c reveals the mean daily temperature at the GST monitoring sites. The lowest GST values (< -10 ℃) occurred in October-December under snow-free or thin snow cover conditions. This is because the insulating snow cover typically occurs in November/December, whereas the snow disappears in May or June. However, at all the sites, significant ground cooling was observed even during the January-March period, despite snow depths typically being sufficient to insulate the ground from air temperature fluctuations. For example, at Galeșu, almost every winter exhibited notable short-term fluctuations in the GST regime. Similar, through less pronounced, patterns were also observed at Judele (e.g., winters of 2013-2014 and 2017-2018), Pietrele (e.g., 2012-2013, 2013-2014, 2014-2015, 2016-2017 and 2021-2022) and Valea Rea (e.g., 2015-2016 and 2019-2020). In a few instances, inverse thermal relationships between ground surface temperature and air temperature were recorded, confirming the presence of internal ventilation through the coarse debris during winter. These thermal anomalies are likely driven by advective heat fluxes that remain active beneath thick snow cover and likely contribute to the cold MAGST observed in the rock glaciers. Usually, during March, ground surface temperatures are relatively stable and are mainly driven by conductive processes. In almost all cases, the corresponding WEqT were below -2 ℃, indicating possible or probable permafrost occurrence (Fig. 13b). WEqT values higher than -2 ℃ occurred only at Pietrele in late winter 2013 and 2014. At Valea Rea and Judele, most WEqT values were between -2 and -3 ℃, whereas at Galeșu, the late winter temperatures were considerably lower.

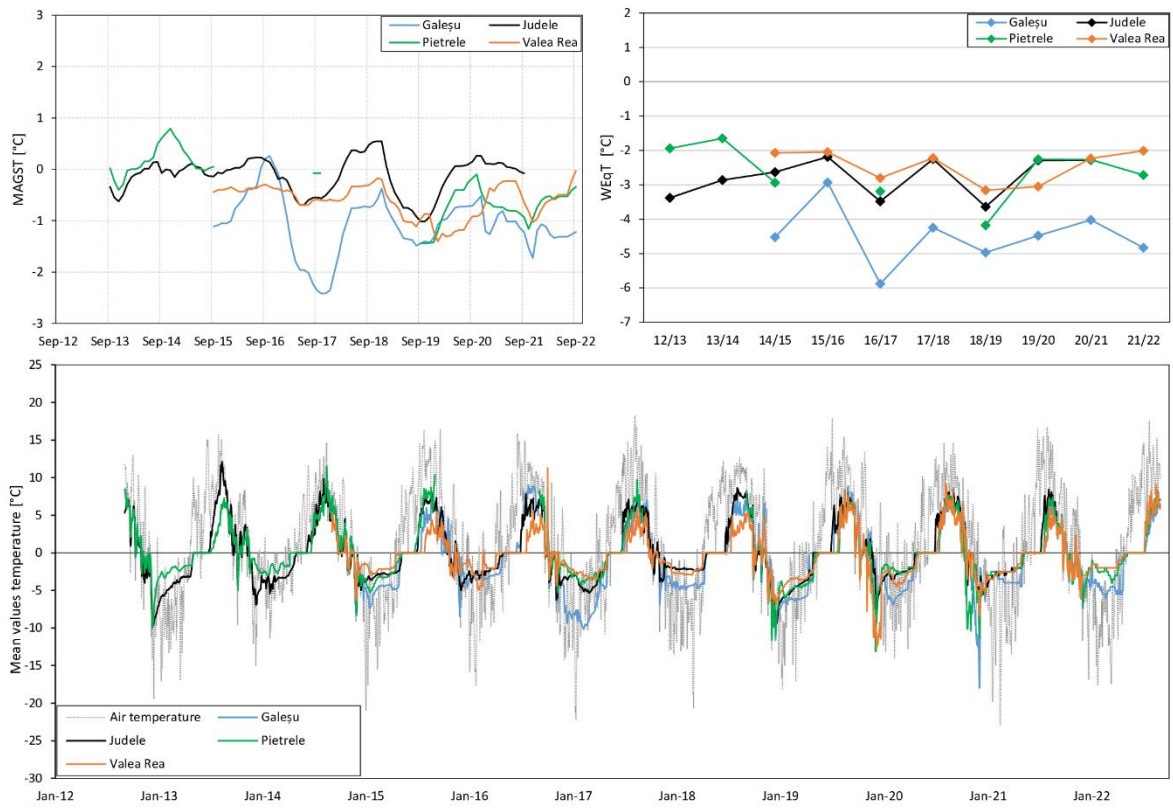


**Figure 13: (a) Running 12-month average of mean annual ground surface temperature, (b) Winter Equilibrium Temperature**
**(WEqT) and (c) Running 12 months average of Ground Surface Temperature (GST) for the period 2012/2013 to 2021/2022 at four**
**sites in the Retezat Mountains and air temperature at Țarcu meteorological station (2180 m).**
**4.4. Geophysics results**
The results of the geophysical surveys at Galeşu rock glacier are shown in Figure 14 for the two crossing ERT profiles P1 and
P2 (Fig. 14a, b), as well as the RST and PJI results available for profile P1 (Fig. 14c, d). Both ERT tomograms reveal an up to
5 m thick layer characterised by high resistivities (> 200 kΩm) representing the dry and coarse-blocky surface layer. Patchy
occurrences with similarly high resistivities are observed in 5-20 m depth in both profiles, which could indicate remnants of
former ice-rich permafrost within the rock glacier (labelled with `ice?` in Fig. 14). Below, a more homogeneous layer of
resistivities around 10 – 30 kΩm may indicate the rock glacier base (bedrock) in the southwestern part of profile P1, however,
the resistivity values do not exclude the possibility of frozen conditions and the interpretations of this zone remains ambiguous.
The landform's overall thickness is estimated at < 20 – 30 m and potentially < 15 m in the southwestern part of profile P1. The
eastern part of profile P1 (x < 50 m in Fig. 14b) is located outside the rock glacier and traverses into a partially vegetated talus
slope. Here, the surface layer exhibits lower resistivities (< 100 kΩm, probably representing smaller block size and organic
material), and the underlying resistive layer (100-200 kΩm) has a thickness of about 10 m and a more homogeneous appearance
than that of the rock glacier. Both the morphology and the resistivity values point to a potentially frozen ventilated talus slope,
but this is not a focus of this study and will not further be explored.

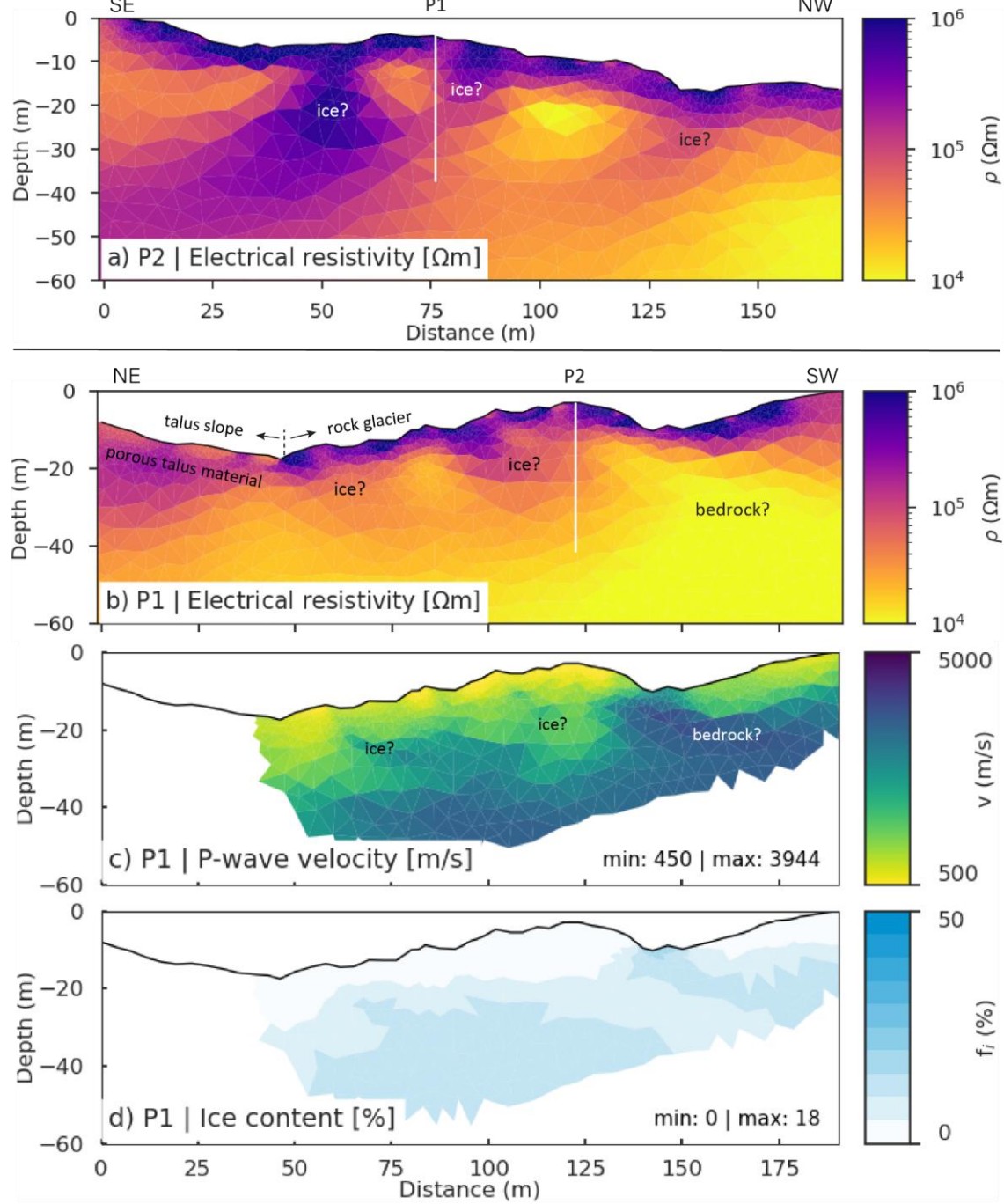


**Figure 14: a) and b) ERT profiles P2 and P1 (see Fig. 1c for location), c) RST profile, and d) modelled ground ice content based on the PJI.**

The seismic profile P1 (Fig. 14c) only covers the rock glacier part of the ERT profile P1 and shows a 5-10 m thick upper layer with P-wave velocities < 1500 m s-1, indicating highly porous blocky material. The velocities are increasing with depth, reaching 4000 m s-1 at about 20-25 m depth in the central part of the rock glacier and at shallower depths (~ 15 m) in the last 50 m of the profile. According to the seismic data, ice-rich permafrost would be possible in large parts of the tomogram. However, in combination with the ERT data, ice-rich conditions seem only plausible in those parts of the tomogram that coincide with elevated resistivities. Alternatively, the zones with lower resistivities could also indicate zones with increased conductivity due to ongoing ice melt, the interpretation therefore remains ambiguous. The seismic data further indicate relatively porous material rather than firm bedrock and point to an overall larger thickness of the rock glacier than indicated by the ERT profile.

The result of the PJI modelling (Fig. 14d) reveals a generally very low ground ice content in the upper sector of the Galeșu rock glacier, with maximum values of 18 %. The highest ice contents coincide with zones with highest seismic velocity, partly contradicting the individual interpretation (bedrock). This can be due to the known rock-ice ambiguity of the current PJI formulation, as the rock and ice content are only directly constrained through the petrophysical equation of the seismic velocity. As a consequence, the correct differentiation between rock and ice is problematic in some cases (see Mollaret et al., 2020 for details). The generally low ice contents modelled through the PJI therefore mainly confirm that a potential former massive ice core is not detectable anymore and point to an advanced state of degradation of this rock glacier. However, these results do not exclude the possibility of more confined ice-saturated or even supersaturated layers (as expected based on the analysis of InSAR data in section). Due to the limited resolution capacity of the geophysical profiles with 4 m sensor spacing thin ice-rich or ice-supersaturated layers may not be resolvable as such, but - depending on their depth and extent - with strongly reduced spatial gradients of the ice content (as well as resistivity or velocity contrast in the individual tomograms). The relatively homogeneous ice content distribution modelled from the PJI may therefore not only reflect the rock-ice ambiguity mentioned above, but also this limitation to resolve small-scale structures.

## 5. Discussion

### 5.1. Assessing the velocity of rock glaciers in marginal periglacial environments

In marginal periglacial regions rock glaciers exhibit a minimal rate of motion (a few cm yr[-1]) (Necsoiu et al., 2016) and evaluating their velocity can pose occasional challenges. Hence, the compilation of MAs inventory might be affected by limitations associated with radar interferometry (Bertone et al., 2022).

The PSInSAR measurements were provided in the SAR LOS direction, representing a 1-D rather than a 3-D measurement, capturing only a single component of motion. Due to the particular steep topography of the study area, it can be assumed that

the movement direction of the actual motion is oriented along the mountain slope. Although PSInSAR does not produce the exact 3-D velocity vectors, this work was helpful in detecting areas of motion and refining the rock glaciers inventory.

Slow-moving areas (i.e., 0.3-1 cm yr$^{-1}$; 1-3 cm yr$^{-1}$) are prevalent in this region, where only 10% of the MAs are characterised by velocities exceeding 3 cm yr$^{-1}$. The latter tend to occupy higher elevations and receive less solar radiation than slower ones. Overall, the median size of the MAs from Retezat Mountains is slightly smaller than other periglacial environments (Bertone et al., 2022). The median size of the MAs, showing minimal variation across velocity classes, exhibits a slight peak among those moving at < 1 cm yr$^{-1}$ (0.33 ha). The median size of MA velocity classes of 1-3 cm yr$^{-1}$ and 3-10 cm yr$^{-1}$ is 0.3 ha, roughly one-third smaller than those reported in Southern Tyrol (Bertone et al., 2024).

The examination of displacement measurements through the differential GNSS technique unveiled similarly very slow movements at specific points (ranging from a few millimetres to 2.8 cm yr$^{-1}$). A comparison between the GNSS survey and PSInSAR results revealed that there was not a very good correspondence between these outcomes. The discrepancy may be due to the difficulty both methods have in accurately detecting such slow movement. Additionally, differences between the velocity datasets may result from the distinct time intervals used for analysis - 2015-2021 for PSInSAR and 2019-2021 for dGNSS. However, most GNSS survey markers that exhibited horizontal displacements exceeding 1 cm yr$^{-1}$ were still located within the MAs. In terms of flow direction, the majority of these markers indicated consistent movement toward the fronts of Judele and Berbecilor rock glaciers, behavior characteristic of permafrost creep.

The PSInSAR analysis enriches the existing rock glacier inventory with information about the activity status of rock glaciers in the Retezat Mountains. A previous study classified 30 rock glaciers in the study area as intact based on geomorphological and ecological criteria (Onaca et al., 2017b). Radar interferometry revealed that 20 rock glaciers exhibit surface displacements exceeding 1 cm yr$^{-1}$. In contrast, 15 rock glaciers showed minimal displacements (0.3 – 1 cm yr$^{-1}$). Of these, eight were previously classified as intact, while the remaining seven were categorised as relict. Four rock glaciers, previously labelled as geomorphologically relict, show surface displacements exceeding 1 cm yr$^{-1}$ and were categorised as transitional in our study. Unlike other regions (e.g., Central Italian Alps, Eastern European Alps, Himalaya) where there is a considerable elevation difference between active/intact and relict rock glaciers (Kellerer-Pirklbauer et al., 2012; Scotti et al., 2013), the Retezat Mountains exhibit a significantly smaller separation.

**5.2. Permafrost occurrence revealed by geophysical and temperature measurements**

The geophysical investigations revealed a frozen layer with overall low ice contents beneath a substantial active layer of approximately 5 m thickness. Similar results have been reported in other marginal periglacial environments (i.e., Făgăraș Mountains, Pirin Mountains, Italian Carnic Alps) where thick active layers indicate even greater thickness (Onaca et al., 2013; Colucci et al., 2019; Onaca et al., 2020). Due to the rock glacier's very dry and extremely coarse blocky surface, the ERT data only have limited quality which is also reflected by the high resistivities of > 200 kΩm on the rock glacier surface. However, we still assume that the overall resistivity pattern indicates the main structures and is representative for the site, whereas small-scale anomalies present in the tomograms as well as absolute resistivity values, should be interpreted with care.

The BTS measurements revealed very cold surface temperatures across large portions of the rock glaciers, with particularly low values observed within MAs, in the upper sections of the rock glaciers, and along the upper talus slopes. However, not all cold areas exhibited detectable movement in the PSInSAR results. Conversely, considerably warmer ground surface temperatures were also recorded in various parts of the rock glaciers, supporting the presence of internal ventilation. The GST results suggest favourable conditions for permafrost existence, but more notably highlight significant ground cooling episodes during the winter, likely driven by air advection. Similar results have been reported for Judele, Pietrele, Valea Rea, Galeşu and Pietricelele rock glaciers in previous studies (Vespremeanu-Stroe et al., 2012; Onaca et al., 2015). Except for Pietrele, the other data-logger sites (Galeşu, Judele and Valea Rea) are located in areas with displacement velocities exceeding 1 cm yr$^{-1}$. However, at Pietrele, despite the prevalence of low BTS values, almost no displacement was observed, except in the western part, where minor displacements (0.3 – 1 cm yr$^{-1}$) were detected. The Pietrele rock glacier is oriented along a south-north direction and the LOS orientation tends to underestimate displacements on north facing slopes (Liu et al., 2013). The lack of significant displacement in this rock glacier between 2014 and 2021 does not necessarily indicate complete ice melt, but likely suggests negligible ice content.

**5.3. Rock glaciers behaviour in marginal periglacial environment**

The rock glaciers in the Southern Carpathians generally move at slower rates than those in other mid-latitude high mountains, where rock glaciers' velocities range from a few centimetres to a few meters per year. But rock glaciers experiencing very low movement velocities were also documented in different periglacial regions (e.g., Pyrenees, Rocky Mountains, northern Norway, Southern Alps of New Zealand, etc.) (Serrano et al., 2010; Brencher et al., 2021; Rouyet et al., 2021; Bertone et al., 2022; Lambiel et al., 2023). In the Retezat Mountains, only 21 % of the inventoried rock glaciers display motion, whereas the rest are considered relict. Similar to the Uinta Mountains (Brencher et al., 2021), in most cases, only a relatively reduced portion of the rock glacier exhibits movements. An illustrative case in this regard is the Galeşu multi-unit rock glacier, displaying movement solely in its uppermost unit, where a younger lobe overlies the main body of the rock glacier. Similar younger lobes were identified in other valleys (e.g., Valea Rea, Pietrele) representing distinct phases of rock glaciers activity, as observed in many other periglacial regions of Europe (e.g. Iceland, European Alps, Cantabrian Mountains, Pyrenees etc.) (Farbrot et al., 2007; Kellerer-Pirklbauer et al., 2008; Steinemann et al., 2020; Amschwand et al., 2021; Oliva et al., 2021; Santos-González et al., 2022).

Additionally, the PSInSAR analysis revealed that, in many instances, the fronts of the rock glaciers in the Retezat Mountains remain stable. Notable examples of stable fronts include Judele (Fig. 11a), Berbecilor (Fig. 11b), Galeşu (Fig. 12a), Păpuşa (Fig. 12c) and Pietricelele (Fig. 12b). Field observations also confirmed that despite steep and sometimes unvegetated slopes, the rock glacier fronts display no recent activity, and no ploughed grass occurs at their snouts. This type of rock glacier, called climatically inactive (Barsch, 1996), is also distinguished by a substantial unfrozen mantle and a low ice content (Onaca et al., 2013).

The geophysical measurements performed in this study indicated the very low ice content in the Galeșu rock glacier, insufficient to support permafrost creep. For permafrost creep to occur, frozen conditions must extend to a depth of at least 10-25 m (Cicoira et al., 2021), which does not appear to be the case at Galeșu. Surface displacements at this site are more likely the result of ice-melt-induced subsidence, solifluction, or the tilting and sliding of blocks within the active layer. In contrast, the flow direction of dGNSS markers at Judele and Berbecilor showed consistent movement patterns, which are not typical of ice-melt subsidence or any other active-layer processes. Additional geophysical and dGNSS measurements are needed to better distinguish between these mechanisms in marginal periglacial environments.

Various studies suggest that the volumetric ice content within active rock glaciers typically falls within the range of 40 % to 60 % (Barsch, 1996; Hausmann et al., 2007; Rangecroft et al., 2015). Conversely, for rock glaciers tending towards inactivity, Wagner et al. (2021) propose average ice content as low as 20 %. These estimates are commonly used to assess the water volume equivalent of ice content stored in rock glaciers (Wagner et al., 2021; Pandey et al., 2024). However, the geophysical investigations presented in this paper reveal even lower values for the volumetric ice content of the Galeșu rock glacier. This finding suggests that the ice content in transitional rock glaciers may be considerably lower than expected, emphasising caution when calculating water volume equivalent on a broad scale.

Considering the current MAGST of approximately -0.5 ℃ and assuming a climatic warming of about +1.5 ℃ since the late 19[th] century (Allen et al., 2018), it is likely that these rock glaciers had a MAGST around -2 ℃ during the pre-industrial period. At such low temperatures, widespread permafrost conditions would have been expected, and the presence of deep permafrost cannot be ruled out. However, accelerated warming in recent decades has resulted in permafrost warming, particularly in ice-poor bedrock, at rates comparable to the increase in air temperature (Noetzli et al., 2024). BTS measurements and GST patterns observed during winter suggest ongoing convective and advective air flow processes that maintain cold ground conditions and support the persistence of ice non-saturated permafrost in the Retezat Mountains.

The results presented in this study align with the coarse-rock glacier hypothesis (Onaca et al., 2015; Popescu et al., 2017), which suggests that permafrost occurrence in the Carpathian Mountains is patchy and limited to sites above 2100 m with low solar radiation. In these locations, very coarse rock glaciers, hosting numerous large boulders, facilitate strong cooling through internal ventilation (e.g., advection and convection) (Wicky and Hauck, 2017; Amschwand et al., 2024) and air stratification (low conductivity) during summer or under thick snow cover.

**6. Conclusions**

This study leads to the following main conclusions:

-	The majority of rock glaciers in the marginal periglacial environment of the Retezat Mountains are classified as relict, with only 21% categorized as transitional. The median elevation of transitional rock glaciers is 150 m higher than that of relict rock glaciers and their median size is slightly smaller. The PSInSAR methodology enabled the identification of new rock glaciers displaying movements, which were initially classified as relict features.

- A total of 92 moving areas were delineated within the rock glaciers of the Retezat Mountains, predominantly falling within the slow-velocity classes (0.3 – 1 cm yr$^{-1}$ and 1-3 cm yr$^{-1}$). Moving areas exhibiting velocities between 1 and 5 cm yr$^{-1}$ are typically located above 2100 m in regions with minimal solar radiation income. Higher movement rates are observed in the upper, younger lobes compared to the well-developed lower parts.

- Long-term ground temperature monitoring from 2012 to 2022 revealed low MAGST values at the observation sites, ranging from -2.3 ℃ to 0.8 ℃. Internal ventilation processes (e.g., advection) occurring throughout the winter significantly contribute to surface cooling and appear to sustain permafrost conditions in coarse debris above 2100 m. This is further supported by BTS measurements, which indicate very cold ground temperatures beneath a thick late-winter snow cover.

- Geophysical measurements conducted on an intact rock-glacier revealed notably low ice content (with maximum values of 18%) in its uppermost section. At this site, surface displacements are most likely driven by processes such as ice-melt-induced subsidence, solifluction or blocks sliding. In contrast, the consistent flow of dGNSS markers towards the fronts of the Judele and Berbecilor rock glaciers points to permafrost creep.

Our findings highlight the value of combining Sentinel-1 SAR data with extensive field investigation (such as DGPS, geophysical and thermal methods) and where possible, with other remote sensing data (like ALOS-2 PALSAR-2), particularly in regions with slow-moving rock glaciers. This approach could serve as a benchmark for similar studies in marginal periglacial environments.

**Code/Data availability**

The deformation data, obtained using PSI, and the temperature data, obtained using GST data loggers and BTS are freely available as a Zendo repository, at https://zenodo.org/records/14544941, DOI: 10.5281/zenodo.14544940
For further questions about data processing readers are encouraged to contact the authors.

**Author Contribution**

The study was conceptualized and managed by AO and FS. AO led the manuscript writing, with contributions from VP, CH, PU, TS and FS. VP, TS, DT, DB and FS contributed to the PSInSAR analysis. FA and IL produced the inventory of moving areas and performed the statistical analysis related to rock glaciers. AO, OB, RP, MV, IL and AVS contributed to the analysis of thermal measurements. CH, BE, SF, RP and AO were involved in conducting and analysing the geophysical measurements. AH and AO provided the GNSS measurements. All authors provided feedback on the final version of the paper.

**Competing interests**

The authors declare that they have no conflict of interest.

**Acknowledgements**

This research was funded by the ESA Permafrost_CCI project (grant number 4000123681/18/I-NB), EEA Norway Grants 2014–2021, under project code RO-NO-2019-0415 / contract no. 30⁄2020 and PNRR-III-C9 2022 - I8, CF 253/29.11.2022, 760055/23.05.2023. We would also like to thank Sabina Calisevici, Adrian C. Ardelean, Patrick Chiroiu, Trond Eiken, Romolus Mălăieștean, Ilie Adrian and Fabian Timofte for support in the fieldwork.

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
