# Peer review of "Slow-moving rock glaciers in marginal periglacial environment of 1"

_EGUsphere, 2024_

## Referee Comment (RC1)

The authors present a range of different observational data related to rock glacier movement, ground surface temperatures, and geophysics from the Carpathian mountains. The work focuses on identifying areas of movement within previously inventoried rock glacier boundaries using PSInSAR. The results are complemented by additional GNSS displacement data from some sites, as well as temperature based investigations of permafrost occurrence and geophysical surveys from individual rock glaciers. The authors show the value of SAR-based displacement mapping for identifying slow movement, which aids activity categorization in rock glacier inventories. The manuscript provides a valuable overview of rock glacier movement in an interesting region with many relict and transitional rock glaciers, where it is of particular importance to work towards understanding ice content and ice content dependent response of movement to climatic forcings.

I find the manuscript to be generally well written. In some sections, the clarity of the text would benefit from restructuring and I had a few questions about methods and results while reading. Color choices in some of the figures could be improved. I am sure my comments and questions can be addressed and confident that this will make a good contribution for ESurf after some manageable revisions mostly pertaining to structure and clarifications in the text.

General comments:
**Methods**: In the section describing the PSInSAR analysis, it would be beneficial to more clearly distinguish between descriptions of the author's approach and EGMS. I feel that the text jumps between both, making it challenging to follow. The Terrasigna method appears to produce substantially improved coverage and this seems like a valuable output, so it would be good to be very clear here about the advantages of the method/product compared to alternatives. See also specific comments below.
**Results:** Some of the figures could be improved regarding colors and visual contrast, see below. I struggle to understand figure 5. This is only mentioned once in the text without much explanation. It would be beneficial to add a few sentences spelling out what is shown in the figure.
**Discussion**: I am unsure why the first part of the discussion (displacement over time) is not in the results section. Unless there is some particular reason for this choice I would suggest moving it to results and restructuring the discussion accordingly.

Language: Ensure consistent use of tenses throughout the manuscript.

Line by line:
**Introduction**
L 45  *increased rock 45 glacier velocities has been observed due to warmer climate*
→ have been
I suggest rephrasing as "..have been linked to warmer climate". You might consider citing Kellerer-Pirklbauer et al 2024 (10.1088/1748-9326/ad25a4) or similar work.

L 62 *transitional rock glaciers behaviour* → rock glacier's behaviour

L85 *At 2000 to 2100 m elevations, the mean annual air temperature hovers around 0° C, with annual precipitations typically around 1000 mm*
This is an awkward sentence construction, consider rephrasing for clarity. I believe "precipitation" should be singular.

Fig 1
Panel a) consider adding the names of the cities to the "city" markers to help readers with orientation.
Panel b) what is the small red dot near rock glacier 6? Does this have any significance?
Panel c) add a scale bar. Consider marking the location of panel c in panel b, e.g. with a box showing the extent of the map as shown in c.

**Methods**
L121 can you specify what the "ecological criteria" were?
L122 *Rock glaciers are categorised into three types based on their activity status: active, transitional, or relict, as outlined by (RGIK, 2023)*

I am unsure whether this paragraph describes what you did or states what the RGIK recommends, or both. In the prior sentence you say kinematics were not used for the inventory since little data is available. However, this sentence reads like you did classify them based on kinematics/activity? Or was this done solely based on geomorphology and ecological criteria? Consider restructuring this part to clarify what you did and how you used your criteria to determine activity status.

L132 Please cite key references for the EGMS products, as listed for example here under "scientific papers"
https://land.copernicus.eu/en/products/european-ground-motion-service?tab=technical_summary

L132 *The algorithms employed in this work closely resemble those described in EGMS Algorithms, with a few notable differences described below*
Clarify what exactly the differences are. You describe what you did in the following paragraphs but do not specify how exactly that is different from EGMS. You are producing a product with improved coverage, so it would be beneficial to state clearly how that is achieved.

L135 *Figure 3b illustrates the coverage of Terrasigna's PSInSAR analysis of the same area, derived in this study solely from one S1 path*
Does the term "Terrasigna's PSInSAR analysis" refer to the analysis you carried out for this study? "Terrasigna" has not been introduced. I understand it is a company but I think this section could be clarified. You might consider something like: We carried out a PSInSAR analysis of the same area, derived solely from one S1 path. In the following, this is referred to as "Terrasigna PSInsSAR".

L139 and following: Similar to the above comment: I think this section could be clarified by restructuring it to more clearly identify your work.

Fig. 3: The legend is hard to read, try to improve image resolution. The yellow-ish colors are hard to distinguish from the true color background image, especially in panel a. Consider an alternative color scale.

L168 *Although permafrost in marginal conditions may occur as patches with a relatively small extent, the minimum size of an MA considered in this study was 300 m2*
Why did you choose this size threshold?

L169 *a one-meter resolution digital elevation model*
Data acquired in which year?

L182 *using a DEM with 10 m pixel spacing*
What is the source of this DEM? Acquisition date? Add citation if possible

**Results**
Fig 4: The green dots are hard to see on the green background. Consider a different color. The relict feature on the right side of the image has several yellow dots and one magenta dot but no MA. Did you apply some kind of minimum dot density to define a MA? Similarly, in the transitional feature in the middle of the image, there is a mix of green and magenta dots in the magenta MA. What was the decision process to classify the MA in the faster category?

Fig 5: This looks like there is a decreasing trend in displacement over time. I assume this is not the correct interpretation, but please clarify by modifying the figure, the caption, and/or adding an explanation in the text. How do the scatter plot and trend lines relate to the inset panel with the map? I assume the MA in the map were derived from the points in the scatter plot - do the MA represent the state of the rock glacier in a particular year? How is negative displacement supposed to be interpreted?

Fig 12: the symbol for "no permafrost" is hard to see, consider using a different color.

Fig 14: Explain the black ellipses in the lowest panel in the caption. Consider adding panel labels that can be referred to in the text. Consider changing the color map in the top two panels (see, for example: https://hess.copernicus.org/articles/25/4549/2021/hess-25-4549-2021.html). I find it hard to distinguish any variation in the lowest panel. Could the scale be changed to make differences more apparent? The highest values on the scale (0.50) do not seem to be reached and the annotation in the plot states max 0.2. Perhaps this would be easier to interpret if the maximum value of the scale was reduced.

**Discussion**
General comment: The first section and Fig 15 seem like results more than discussion. Is there some particular reason why Fig 15 is considered part of the discussion rather than a result? The linkage to climate drivers is suitable for the discussion but consider placing the analysis of movement over time in the results.

L 383: *"..exhibited consistent movement between 2016 and 2021 and two distinct types of velocity were identified.."*
Please define what you mean by "consistent movement" and explain how "distinct types" of velocity were determined.

Fig 15: what are the black lines? Explain in the caption. The text states that "two distinct types of velocity were identified" but does not clarify how.

L436: *Additionally, the PSInSAR analysis revealed that, in many instances, the fronts of the rock glaciers in the Retezat Mountains remain stable*
Is this evident in any of the figures? Is it possible to show an example?

L448 Interesting!

L455: consider adding some references to work related to convective cooling in coarse blocks. For example: https://doi.org/10.5194/tc-18-2103-2024 https://doi.org/10.5194/tc-11-1311-2017 or anything else you find pertinent.

---

## Author Response (AR1)

**RC1: 'Comment on egusphere-2024-3262', Anonymous Referee #1, 14 Mar 2025**

**Author comments (AC) in blue**

The authors present a range of different observational data related to rock glacier movement,ground surface temperatures, and geophysics from the Carpathian mountains. The workfocuses on identifying areas of movement within previously inventoried rock glacier boundaries using PSInSAR. The results are complemented by additional GNSS displacement data from some sites, as well as temperature based investigations of permafrost occurrence and geophysical surveys from individual rock glaciers. The authors show the value of SAR-based displacement mapping for identifying slow movement, which aids activity categorization in rock glacier inventories. The manuscript provides a valuable overview of rock glacier movement in an interesting region with many relict and transitional rock glaciers, where it is of particular importance to work towards understanding ice content and ice content dependent response of movement to climatic forcings. I find the manuscript to be generally well written. In some sections, the clarity of the text would benefit from restructuring and I had a few questions about methods and results while reading. Color choices in some of the figures could be improved. I am sure my comments and questions can be addressed and confident that this will make a good contribution for ESurf after some manageable revisions mostly pertaining to structure and clarifications in the text.

Thank you for your thorough review and insightful remarks, which have significantly contributed to improving the clarity and quality of the manuscript. We hope that our revisions adequately address your concerns. Your comments have added considerable value to our research, and we appreciate the opportunity to refine the paper accordingly. Our detailed responses to each comment are provided below.

**General comments:**

Methods: In the section describing the PSInSAR analysis, it would be beneficial to more clearly distinguish between descriptions of the author's approach and EGMS. I feel that the text jumps between both, making it challenging to follow. The Terrasigna method appears to produce substantially improved coverage and this seems like a valuable output, so it would be good to be very clear here about the advantages of the method/product compared to alternatives. See also specific comments below.

Thanks for this comment. We made the text clearer and added a detailed description of the Terrasigna method. `The PSInSAR analysis employed in this study was developed by Terrasigna and generally follows the EGMS specifications (https://land.copernicus.eu/en/technical-library/egms-algorithm-theoretical-basis-document/@@download/file). However, there are a few differences, particularly related to the choice of reference points, the modelling of atmospheric effects in steep terrain and the selection of SAR images. EGMS products are computed at the regional level, where reference points are typically located in lowland areas covered by infrastructure, which provides strong and stable radar backscattering. Additionally, EGMS includes all available acquisitions, even those affected by snow cover at high altitudes. However, inspection of EGMS products reveals that extending the measurement network from lowland reference points to mountain summits was largely unsuccessful. This is mainly because the atmospheric path delay associated with steep topography was not adequately compensated for and acts like phase noise. Furthermore, snow cover during winter significantly impacts data quality. Non-homogeneous snow or snow with variable humidity scatters radar signals and increases phase noise. If the case of dry snow, radar waves penetrate the snowpack, but because they propagate more slowly than in air, the interferometric phase experience a time delay. This delay produces apparent subsidence (false ground*

*displacement away from the radar sensor). Combined these factors often result in the rejection of radar targets on mountain tops due to excessive noise.*

*To address these issues, Terrasigna carefully selected reference points located on mountain summits, where the topographic atmospheric delay is similar to that of the areas of interest. Additional efforts were made to improve the accuracy of atmospheric delay modelling and compensation. Furthermore, only snow-free acquisitions were considered. As a result, the high density of radar targets – formed by bare rocks at the top of the mountains – is preserved in our measurements. `*

Results: Some of the figures could be improved regarding colors and visual contrast, see below. I struggle to understand figure 5. This is only mentioned once in the text without much explanation. It would be beneficial to add a few sentences spelling out what is shown in the figure.

We have changed most of the figures and improved visibility according to reviewer`s suggestions. Also, we changed some colored polygons to hashed polygons to make the figures easier to read. We discuss fig.5 further down.

Discussion: I am unsure why the first part of the discussion (displacement over time) is not in the results section. Unless there is some particular reason for this choice I would suggest moving it to results and restructuring the discussion accordingly.

Thanks for this comment. Indeed, this part belongs to the Results. However, following suggestions of Reviewer 2 we deleted this Figure and corresponding interpretation.

**Language:** Ensure consistent use of tenses throughout the manuscript.

Thanks for this comment. The manuscript has been revised for consistent past tense usage.

**Line by line:**

**Introduction**

L 45 increased rock 45 glacier velocities has been observed due to warmer climate → have been I suggest rephrasing as "..have been linked to warmer climate". You might consider citing Kellerer-Pirklbauer et al 2024 (10.1088/1748-9326/ad25a4) or similar work.

We reformulated according to your suggestion and added the recommended reference.

L 62 transitional rock glaciers behaviour → rock glacier's behaviour

Following suggestions of Reviewer 2, Introduction was considerably modified and this phrase was deleted.

L85 At 2000 to 2100 m elevations, the mean annual air temperature hovers around 0° C, with annual precipitations typically around 1000 mm. This is an awkward sentence construction, consider rephrasing for clarity. I believe "precipitation" should be singular.

We reformulated according to your suggestion.

Fig 1 Panel a) consider adding the names of the cities to the "city" markers to help readers with orientation. Panel b) what is the small red dot near rock glacier 6? Does this have any significance? Panel c) add a scale bar. Consider marking the location of panel c in panel b, e.g. with a box showing the extent of the map as shown in c.

Based on suggestions from both reviewers, we modified Fig.1 as follows:

Panel (a) We added the names of the cities and we changed the background to a hillshade, for better readability.

Panel (b) The red dot was a cartographic mistake and we removed it. We changed the content of the legend, from "permafrost extent" to "Modelled permafrost extent (Popescu et al. 2024)" and from "Differential DGPS survey to "Differential GNSS survey". We marked the location of panel (c)

Panel (c) We added a scale bar. We used an updated version of the rock glacier inventory in Retezat, with both extended and restricted outlines, and with better delimitation between RG and talus. The panel has a northern orientation, in line with the other two panels. We changed the symbols and annotations for the geophysical profiles and we zoomed in the new rock glacier outline, making the profiles more visible.

**Methods**

L121 can you specify what the "ecological criteria" were?

We added the following text: `(e.g., degree and type of vegetation cover). `

L122 Rock glaciers are categorised into three types based on their activity status: active, transitional, or relict, as outlined by (RGIK, 2023). I am unsure whether this paragraph describes what you did or states what the RGIK recommends, or both. In the prior sentence you say kinematics were not used for the inventory since little data is available. However, this sentence reads like you did classify them based on kinematics/activity? Or was this done solely based on geomorphology and ecological criteria? Consider restructuring this part to clarify what you did and how you used your criteria to determine activity status.

We have rearranged the paragraphs to improve clarity. Additionally, we now use only the definitions provided by RGIK (2013) for active, transitional, and relict rock glaciers, and have cited the source accordingly.

L132 Please cite key references for the EGMS products, as listed for example here under "scientific papers" https://land.copernicus.eu/en/products/european-ground-motion-service?tab=technical_summary

We added the recommended citation.

L132 The algorithms employed in this work closely resemble those described in EGMS Algorithms, with a few notable differences described below. Clarify what exactly the differences are. You describe what you did in the following paragraphs but do not specify how exactly that is different from EGMS. You are producing a product with improved coverage, so it would be beneficial to state clearly how that is achieved.

Thank you for this comment. We have added a more detailed explanation in the `General comments – Methods` section, outlining the difference between EGMS Algorithms and the Terrasigna method, and

explaining the rationale behind using the Terrasigna product. Please refer to our response above under `General comments`.

L135 Figure 3b illustrates the coverage of Terrasigna's PSInSAR analysis of the same area, derived in this study solely from one S1 path. Does the term "Terrasigna's PSInSAR analysis" refer to the analysis you carried out for this study? "Terrasigna" has not been introduced. I understand it is a company but I think this section could be clarified. You might consider something like: We carried out a PSInSAR analysis of the same area, derived solely from one S1 path. In the following, this is referred to as "Terrasigna PSInsSAR".

Thank you for this comment. We have changed accordingly.

L139 and following: Similar to the above comment: I think this section could be clarified by restructuring it to more clearly identify your work.

Thank you for this comment. We have rephrased this section. We added the following phrases: `It is evident that the EGMS coverage is relatively sparse and does not highlight dynamic areas—there are no zones marked in red, which typically indicate significant ground motion. In contrast, the PSInSAR results from Terrasigna show much denser coverage and clearly identify dynamic areas, with red colors representing higher displacement rates. `

Fig. 3: The legend is hard to read, try to improve image resolution. The yellow-ish colors are hard to distinguish from the true color background image, especially in panel a. Consider an alternative color scale.

Thank you for this comment. Based on suggestions from both reviewers, we modified Fig.3 as follows: we zoomed in to the central area of Retezat Mountains and we changed the background to a hillshade, for better visibility. We also enlarged the legend to be easier to read.

L168 Although permafrost in marginal conditions may occur as patches with a relatively small extent, the minimum size of an MA considered in this study was 300 m2 Why did you choose this size threshold?

Thank you for this comment. We removed this phrase, as there is currently no consensus in the literature on this matter. For instance, Bertonne et al. (2024) used 20 Sentinel-1 pixels (with a resolution of 5–20 m), corresponding to an area of approximately 100–400 m², while in Bertonne et al. (2022), the analysis was based on 20–30 pixels. However, in response to Reviewer 2's suggestion, we have increased the minimum area threshold for MAs to 1000 m² to better capture the signal of permafrost creep.

L169 a one-meter resolution digital elevation model Data acquired in which year?

We have added the year of data acquisition for Lidar data.

L182 using a DEM with 10 m pixel spacing. What is the source of this DEM? Acquisition date? Add citation if possible

The source of this DEM is 1:25 000 scale topographic maps with a contour interval of 10 m.

**Results**

Fig 4: The green dots are hard to see on the green background. Consider a different color. The relict feature on the right side of the image has several yellow dots and one magenta dot but no MA. Did you apply some kind of minimum dot density to define a MA? Similarly, in the transitional feature in the middle of the image, there is a mix of green and magenta dots in the magenta MA. What was the decision process to classify the MA in the faster category?

Thank you for this comment. Based on suggestions from both reviewers, we modified Fig. 4 as follows: we zoomed in on the figure and we only presented two rock glaciers, one relict and one transitional. In order to be able to have the same color for corresponding velocity classes for the MA and PSI dots we choose to represent the MA with hashed polygons, instead of colored polygons.

Fig 5: This looks like there is a decreasing trend in displacement over time. I assume this is not the correct interpretation, but please clarify by modifying the figure, the caption, and/or adding an explanation in the text. How do the scatter plot and trend lines relate to the inset panel with the map? I assume the MA in the map were derived from the points in the scatter plot - do the MA represent the state of the rock glacier in a particular year? How is negative displacement supposed to be interpreted?

We added the following caption to fig.5:

Displacement profiles over a period of 6 years (2015 – 2021) for 4 locations (identified in the location map with dots of corresponding color) representing the three moving areas and one for an area with no movement. The dots show the actual PSI displacement measurements, while trend lines (linear regressions) indicate long-term motion patterns. The displacement is measured relative to 2021. The downward trend can be interpreted as movement away from the sensor, which in our case represents a combination of vertical movement and horizontal downslope movement. The gap in point density along the trend line is due to the winter season, the measurements being performed in snow free conditions, usually from June to November.

Further explanation of the curves: Each dot represents the displacement between its corresponding date on the X axis and the reference point in 2020. For the magenta line the displacement is approximately 20cm (from 17 in 2015 to -3 in 2021) which over a period of 6 years averages to about 3.3 cm yr $^{-1}$(note that the sign of the displacement value represents the displacement direction compared to 2020). The trend line shows the direction of the movement (away from sensor) and it also shows that the movement is consistent over the study period. Some seasonal displacement trends can be observed (i.e. the displacement during the winter observed in the magenta line or the lack of displacement during some summer periods) but these trends are not consistent in the whole study area. Also tracking seasonal variations presents high uncertainty, thus we prefer to present only annual or multiannual displacement rates.

Fig 12: the symbol for "no permafrost" is hard to see, consider using a different color.

Thank you for this comment. We revised Figure 12 in response to Reviewer 2's suggestions, removing the 'no permafrost' class and presenting BTS values without inferring permafrost extent. To address concerns about visibility, we now represent MAs using hashed polygons instead of colored ones, making the BTS data points more distinguishable.

Fig 14: Explain the black ellipses in the lowest panel in the caption. Consider adding panel labels that can be referred to in the text. Consider changing the color map in the top two panels (see, for example: https://hess.copernicus.org/articles/25/4549/2021/hess-25-4549-2021.html). I find it hard to distinguish

any variation in the lowest panel. Could the scale be changed to make differences more apparent? The highest values on the scale (0.50) do not seem to be reached and the annotation in the plot states max 0.2. Perhaps this would be easier to interpret if the maximum value of the scale was reduced.

Thank you for this comment. We adjusted the panel labels to highlight more prominently the profile number and parameter (electric resistivity, P-wave velocity, or ice content). We also changed the color map of the ERT tomograms to the plasma color map, which is equivalent to the viridis color map used for the seismic tomogram and has a clear gradient from light to dark, improving readability for color-blind persons. Further, the color map of the ice content tomogram was adjusted to discrete intervals of 5 %, which hopefully improves the readability of this panel.

The black ellipses were removed from the figure, as they are not needed anymore based on the changes above and in combination with the modifications in response to reviewer 2 leading to revisions of the overall interpretation.

**Discussion**

General comment: The first section and Fig 15 seem like results more than discussion. Is there some particular reason why Fig 15 is considered part of the discussion rather than a result? The linkage to climate drivers is suitable for the discussion but consider placing the analysis of movement over time in the results.

We have removed this section in response to Reviewer 2's suggestion.

L 383: "..exhibited consistent movement between 2016 and 2021 and two distinct types of velocity were identified.." Please define what you mean by "consistent movement" and explain how "distinct types" of velocity were determined.

We have removed this section in response to Reviewer 2's suggestion.

Fig 15: what are the black lines? Explain in the caption. The text states that "two distinct types of velocity were identified" but does not clarify how.

We have removed this figure in response to Reviewer 2's suggestion.

L436: Additionally, the PSInSAR analysis revealed that, in many instances, the fronts of the rock glaciers in the Retezat Mountains remain stable Is this evident in any of the figures? Is it possible to show an example?

Thank you for this comment. We have added: `*Notable examples of stable fronts include Judele (Fig. 11a), Berbecilor (Fig. 11b), Galeșu (Fig. 12a), Păpușa (Fig. 12c) and Pietricelele (Fig. 12b).* `

L448 Interesting!

Thanks for this comment.

L455: consider adding some references to work related to convective cooling in coarse blocks. For example: https://doi.org/10.5194/tc-18-2103-2024 https://doi.org/10.5194/tc-11-1311-2017 or anything else you find pertinent.

We added the recommended references.

**RC2: 'Comment on egusphere-2024-3262', Anonymous Referee #2, 07 Apr 2025**

**General comment**

Very interesting and pertinent multi-method study which intends to contribute filling knowledge gaps on various structural, thermal and kinematic aspects of rock glaciers, whose current activity state, meaning their ability to transfer sediment downslope, is ranging from relict (no movement) to transitional (few localized or widespread movement).

In its current form, the paper suffers however several weaknesses.

There is a frequent conceptual mixing between landforms (rock glaciers), processes (permafrost / rock glacier creep), ground thermal state (permafrost conditions or not), ground ice occurrence at depth and kinematics, which prevents a clear understanding of the paper content (data interpretation). This must be disentangled.

There is often, but not everywhere, some lack of precision/clarity in the text (is the purpose fully clear and understandable for the reader ?).

There are several questions arising about the applied methods and the interpretation of their results.

The illustrations must be improved.

In general, the paper could be very worth of being published in ESurf. However, in its current form, it requires major improvements.

My comments are going sometimes pretty far into details, when not too much. This must be understood as a constructive contribution from my side in order for the authors to improve the pertinence of their paper.

**Response**

We sincerely appreciate your careful review and valuable feedback on our manuscript. Your insightful comments have played a crucial role in enhancing the clarity and quality of our study. We have thoughtfully considered all your suggestions and have incorporated the necessary revisions into the updated manuscript.

Below, we provide detailed explanations of the changes made in response to each of your comments, with line references corresponding to the version with accepted modifications. We hope our revisions effectively address your concerns and further strengthen the manuscript.

Thank you once again for your time and thoughtful review.

**Detailed comments**

**Abstract**

Not fully supported by data. For instance, (L. 29) "The slow surface movement of rock glaciers in the marginal periglacial mountains is driven by the deformation of thin, frozen layers". First, the type of movement, which appears to occur in some parts of some of the studied rock glaciers in the Retezat mountains, has not been explained (permafrost creep ? vertical subsidence ? other cause ?). Second, geophysical investigation has only been conducted on a reduced section of one single rock glacier, which prevents any generalization. Third, the statement appears to never being discussed in the paper (or I missed it). Finally, the authors' statement cannot be extended to all "marginal periglacial mountains" on the basis of this single study.

This is just one example and an invitation to revise/verify the entire abstract once the paper will be improved.

The last sentence about "the benefit of combining Sentinel-1 SAR data with comprehensive field investigations" is common sense and out of the scope of the paper (-> to be removed).

Thanks for your comment. We have removed the phrase and revised the abstract to reflect the updated version of the manuscript.

**Main text**

L.35. "In high mountains, ice-rich permafrost occurrence is usually associated with rock glaciers". Rock glacier is a feature of the mountain periglacial landscape. It is not a question of the "highness" of the mountain, but of the occurrence of periglacial conditions, whose lowermost elevation on a mountain range is very roughly related to latitude and overall climate conditions. Ice-rich permafrost is an obscure concept that must be clearly defined by the authors (what do they mean here ? ice-oversaturation ? >30, 50 or 70% of the total volume ?). There are many ice-rich (almost whatever the definition) permafrost occurrences outside of rock glaciers (e.g. moraines, talus, landslides). Ice-rich permafrost is unlikely to occur in relict rock glaciers, but permafrost occurrence cannot be excluded. I imagine that the authors would like to say that the process conducting to the development of rock glacier (permafrost / rock glacier creep) is usually associated with the occurrence of saturated to super-saturated ground ice conditions (ice-rich) on mountain slopes.

Thank you for your valuable comments. The Introduction has been substantially revised in accordance with your suggestions. We have removed the first phrase and replace it with `*Rock glaciers are prominent landforms in the periglacial environment, serving as indicators of permafrost presence at the time of their formation (Haeberli et al., 2006).* `

L.35-36 "Rock glaciers are masses of debris-ice mixture common in many cold mountains on Earth". A rock glacier is basically a mass of debris (see RGIK (2023) definition), which undergoes or underwent deformation (permafrost creep). There is no ice (or only few, unsaturated ground ice) in relict rock glaciers. Cold does not mean anything clear. The sentence must be adapted accordingly.

We have replaced the second phrase with: `*Generated through past or present permafrost creep, they are debris-dominated features typically identified by their front, lateral margins and occasionally ridge-and-furrow surface patterns (RGIK, 2023).* `

L.36-37. When occurring (not all rock glaciers comprise it), "the coarse debris surface of rock favors ground cooling". Agreed. It produces a so-called thermal offset. However, why should it decisively contributes to preserving permafrost over long periods ? The thermal offset effect is not expected to change if for instance the temperature is warming. So, what would the authors precisely say? I imagine that they would mean that the rock glaciers in the Southern Carpathians should have formed under colder pre-holocene (?) climate conditions, but that their coarse surface debris has contributed to preserve cold enough ground conditions for permafrost to subsist at least locally? Harris and Pedersen (1998) is maybe not the most relevant reference.

We deleted these two phrases and replaced them with a new paragraph that addresses the comment you provided. `In the Southern Carpathians, rock glaciers mostly present as relict landforms, yet retain isolated permafrost in certain areas (Vespremeanu-Stroe et al., 2012; Onaca et al., 2013, 2015; Popescu et al., 2015, 2024). Indicators such as extensive lichen cover, vegetated fronts and the overall morphological stability of many landforms suggest that permafrost creep is significantly reduced compared to the colder climatic conditions of the pre-Holocene (Popescu et al., 2017). These rock glaciers are predominantly mantled by angular, coarse-grained blocks which facilitate ground cooling (Onaca et al., 2017). The thermal offset associated with this blocky surface layer contributes to the maintenance of subzero temperatures in the subsurface over prolonged periods (Kellerer-Pirklbauer, 2019), thereby enhancing permafrost preservation even at relatively low altitudes (Colucci et al., 2019). In addition, the `chimney effect` - an advective heat flux process (Delaloye and Lambiel, 2005) - significantly contributes to surface cooling in highly porous, openwork structures. `

L.38. What is the "themal inertia" of the thick coarse debris layer, which, as said earlier, is not always thick ? What does "thick" mean ?

We deleted this phrase.

L.39. Permafrost is not resilient to climate change. It is just thermally responding with delay due to the time necessary to diffuse the surface temperature forcing at depth and the large consumption (when warming) of heat by the phase change of ice to water.

We deleted this phrase.

L.40. What is defining the regional limit of permafrost ? Many people have used rock glaciers to do it... Moreover, what is the significance of such geomorphologically-based limit under current changing climate conditions ?

We deleted this phrase.

Aside from the surface debris, the occurrence of permafrost conditions at "low" elevation as in relict rock glaciers is possibly due to the internal (advective) ventilation of the rock glacier. Some of the authors have worked on such a process. Maybe worth of adding a few words on it.

Thank you for suggestion. We added this phrase `In addition, the `chimney effect` - an advective heat flux process (Delaloye and Lambiel, 2005) - significantly contributes to surface cooling in highly porous, openwork structures.`

The climate is warming. In Central Europe, current MAAT could be up to 2°C warmer than a few decades ago, meaning an approximative rise of the isotherms by 400 m in elevation. One should now be very

careful when referring to any MAAT value. The current permafrost occurrences are obviously not defined by the current MAAT, but still in most cases by the climate conditions of the previous decades/centuries.

Thank you for your suggestion. We have deleted the phrase referring to climate warming and permafrost sensitivity, but we presented this idea in the Discussion section.

At the end of this first paragraph, I don't always understand the link between the various sentences and I don't see where the authors would like to go. They should first state that their studied region is bearing many rock glaciers usually exposing a coarse blocky surface. According to the stability of their (vegetated) fronts and their subdued morphology, these landforms are mostly considered as relict, meaning having developed under colder conditions than those prevailing during the last centuries/millenia. However, the coarse grained surface and its cooling effect (say how) is expected to have preserved permafrost conditions in more or less large areas of these rock glaciers, in particular in their uppermost sections.

Thank you for your suggestion. We have reorganized the Introduction in accordance with your suggestions. We have addressed the specific characteristics of rock glaciers in the Southern Carpathians. See one of my previous comments.

2nd paragraph

This paragraph is focusing on rock glacier kinematics (movement). There is however no word about the mechanism (creep) making the rock glacier to develop (have developed), as well as about further processes which could contribute to a tiny deformation of the surface to occur independently of the permafrost creep process (e.g. ice-melt induced subsidence, solifluction of the active layer).

Thanks for your comment. We have presented the mechanisms that may be responsible for surface displacements of rock glaciers. We added the following phrases: `Permafrost creep encompasses both the internal deformation of ice within the frozen material and shearing at discrete planes within or just beneath the frozen structure (Cicoira et al., 2021). Surface displacement can also result from processes occurring within the active layer, such as ice-melt-induced subsidence, solifluction, or the tilting and sliding of blocks, which may act independently of or in addition to permafrost creep (Serrano et al., 2010; Cicoira et al., 2021). `

In addition to that :

L. 43. "The surface kinematics of rock glaciers have garnered significant interest from the international community in recent years (Bearzot et al. 2022)…". Right, but the reference is not suited. This is just one study on one single rock glacier. Prefer references like Kellerer-Pirklbauer et al. (2024) (over the entire Alps), Hu et al. (2025) (review paper on rock glacier velocity as a climate indicator – okay, it was published after the submission of the present manuscript), Pellet et al. (2024) (BAMS report, worldwide) and/or other references therein (e.g. Kääb and Røste 2024 about rock glaciers in the U.S.)

Thank you for recommending appropriate references. We added all here.

L. 45. Explain why rock glaciers are moving faster under warmer conditions. Does this statement also work for very slow moving rock glaciers (< 3 cm/y) ?

We added the following phrases: `*Rising temperatures within frozen debris enhance movement rates, as warming reduces the viscosity of the ice and promotes additional lubrification from infiltrating water (Kääb*

*et al., 2007). According to Necsoiu et al. (2016), slow-moving rock glaciers in the Southern Carpathians exhibited increased velocities between 2007 and 2014, attributed to rising permafrost temperatures. `*

L. 47. "… occasional accelerations… (Vivero et al. 2022) ». What is the meaning of "occasional" and why "accelerations" (plural) ? Why not to speak about destabilization and refer to other various related papers (e.g. Roer et al. 2008, Delaloye et al. 2013, Eriksen et al. 2018, Marcer et al. 2021, Hartl et al. 2023). The study by Vivero et al. 2022 is first and mostly a technical paper, referring to a single rock glacier, suffering a single destabilization phase (its first one in this amplitude) since a decade or so.

Thank you for your suggestions. We replaced "occasional accelerations" with `destabilization` and deleted the citation of Vivero et al., 2020. We also referred to the papers recommended by you (e.g. Roer et al. 2008, Delaloye et al. 2013, Eriksen et al. 2018, Marcer et al. 2021, Hartl et al. 2023).

L. 48. What is this sentence about permafrost warming doing here (section on rock glacier kinematics) ? What is meant by "increased sensitivity" ? Maybe worth of referring also to Noetzli et al. 2024.

We deleted this phrase.

L.54. The "ice content in this category of rock glaciers" (transitional) is not known. That it is "below a certain saturation threshold" is only presumed. In such a case the shear strength (internal friction) is too high to favor fast creep movement, but not the "shear stress is too weak". Is there not something younger than Barsch (1996) as a reference ? E.g. Cicoira et al. 2020 or references therein ?

We deleted this phrase and replaced with `*This reduced surface velocity is attributed to the high shear strength of the material, which inhibits fast creep movement (Cicoira et al., 2021).*`

I stop with the detailed commenting here, but it gives an idea about the improvement the manuscript could need in its precision.

Thank you. We have improved the overall accuracy and clarity of the text throughout the entire manuscript.

L. 59-61**.** The expected relationship between surface movement, permafrost occurrence and areas bearing ground ice must be explained. Consistent surface movement (consistent/homogeneous horizontal flow field, not dominated by subsidence) could relate to permafrost creep (the latter must be explained before; note that frozen conditions down to a minimum of about 20 m are necessary for permafrost creep to occur), which point out to the occurrence of thick "ice-rich" permafrost. Isolated or thinner ice-rich permafrost cannot produce rock glacier creep, whereas the latter is also not occurring in every permafrost area.

We added the following phrases: `*Surface displacements can be attributed to permafrost creep only if the flow direction and velocity remain spatially consistent and uniform over a documented period (RGIK, 2023). Permafrost creep typically occurs when the thickness of the ice-rich core in rock glaciers reaches at least 10-25 m (Cicoira et al., 2021). In contrast, displacements observed in rock glaciers with thinner layers of frozen debris are primarily driven by deformations within the active layer above the permafrost table. `*

L. 72. Express the value of the "significant warming that the Southern Carpathians are facing". It will be very important for the interpretation of the current ground surface temperatures.

We added the following phrase: `*Above 2000 m in the Southern Carpathians, the 1991-2020 climatological period was 0.8 ºC warmer than the 1961-1990 baseline (Berzescu et al., 2025).* `

L. 85. MAAT must be dated (because of the ongoing significant warming". MAAT = 0°C at 2000-2100 m a.s.l. is particularly low according to the apparent elevation of the transitional rock glaciers. Provide also elevations for the latter (or refer the adequate section).

We have dated it `*(calculated for the period 1961-2007)* `.

L.92. Rock glaciers have developed on granites (the subjacent bedrock is granite) or have they developed from granite (the bedrock in the source area is granite) ?

We added `*granite bedrock*`.

L.105. It would be worth of precising how the absence of glacier development during the Younger Dryas has been stated (model ?  lack of geomorphological evidence ?). It is important as it provides an idea about when the rock glaciers started to develop ? So, the rock glaciers located in the uppermost valley sections could have already developed during the Younger Dryas or even before and for most of them have become relict or almost relict since the beginning of the Holocene about 10'000 years ago.

We added the following text `*…based on cosmic-ray exposure dating (Ruszkiczay-Rüdiger et al., 2021). Rock glaciers in the Retezat Mountains likely began to develop during the Younger Dryas, with most having become relict or transitional since the onset of the Holocene.* `

Figure 1 :

- Better to provide a hillshade background and the name of the highlighted cities.

- Write in the map legend " Modelled permafrost extent" or better "Modelled permafrost extent (Popescu et al. 2024)"; Differential GNSS survey or dGNSS survey

- It is a detail, but it would be nice to keep the same orientation on b).

What is in red ? The rock glacier outline I presume. It would be good to apply the RGIK guidelines to first provide both the extended and restricted outlines and to also remove the feeding talus slopes from the rock glacier outline.

Just write P1 and P2 and do it larger.

Based on suggestions from both reviewers, we modified Fig.1 as follows:

Panel (a) We added the names of the cities and we changed the background to a hillshade, for better readability.

Panel (b) The red dot was a cartographic mistake and we removed it. We changed the content of the legend, from "permafrost extent" to "Modelled permafrost extent (Popescu et al. 2024)" and from "Differential DGPS survey to "Differential GNSS survey". We marked the location of panel (c)

Panel (c) We added a scale bar. We used an updated version of the rock glacier inventory in Retezat, with both extended and restricted outlines, and with better delimitation between RG and talus. The panel has a northern orientation, in line with the other two panels. We changed the symbols and annotations for the geophysical profiles and we zoomed in the new rock glacier outline, making the profiles more visible.

L. 122-125 If you refer to RGIK (2023), it must be fully respected (not changing the definition), done for the entire paragraph and not mixed with Barsch (1996).

We corrected the existing phrases and respected the definition by RGIK (2023): *`Active rock glaciers exhibit consistent downslope movement across most of their surface with displacement rates ranging from a decimeter to several meters per year (RGIK, 2023). Most of the surface of a transitional rock glacier experiences little to no downslope movement, with annual average displacement rates generally falling below one decimeter (RGIK, 2023). Rock glaciers exhibiting no detectable movement across most of their surface are classified as relict (RGIK, 2023). `We are not referring to the study by Barsch (1996) here.*

L.132. "…millimetric accuracies, similar to GNSS": to dGNSS ? The accuracy of the latter is nevertheless not millimetric, but centimetric

We deleted GNSS.

L. 135-138. What is Terrasigna's PSInSAR ? Why is it providing much more data points than EGMS ?

We made the text clearer and added a detailed description of the Terrasigna method. *`The PSInSAR analysis employed in this study was developed by Terrasigna and generally follows the EGMS specifications (https://land.copernicus.eu/en/technical-library/egms-algorithm-theoretical-basis-document/@@download/file). However, there are a few differences, particularly related to the choice of reference points, the modelling of atmospheric effects in steep terrain and the selection of SAR images. EGMS products are computed at the regional level, where reference points are typically located in lowland areas covered by infrastructure, which provides strong and stable radar backscattering. Additionally, EGMS includes all available acquisitions, even those affected by snow cover at high altitudes. However, inspection of EGMS products reveals that extending the measurement network from lowland reference points to mountain summits was largely unsuccessful. This is mainly because the atmospheric path delay associated with steep topography was not adequately compensated for and acts like phase noise. Furthermore, snow cover during winter significantly impacts data quality. Non-homogeneous snow or snow with variable humidity scatters radar signals and increases phase noise. If the case of dry snow, radar waves penetrate the snowpack, but because they propagate more slowly than in air, the interferometric phase experience a time delay. This delay produces apparent subsidence (false ground displacement away from the radar sensor). Combined these factors often result in the rejection of radar targets on mountain tops due to excessive noise.*

*To address these issues, Terrasigna carefully selected reference points located on mountain summits, where the topographic atmospheric delay is similar to that of the areas of interest. Additional efforts were made to improve the accuracy of atmospheric delay modelling and compensation. Furthermore, only snow-free acquisitions were considered. As a result, the high density of radar targets – formed by bare rocks at the top of the mountains – is preserved in our measurements. `*

Why is it only using one path (descending) ?

We added the following phrases: *`Both ascending and descending paths were processed for cross-validation, along with L-band ALOS data, which was analyzed for the same purpose. Because descending passes occur in the early morning—when atmospheric conditions are generally more stable than in the evening—the resulting measurements tend to be less noisy. A 2D decomposition between ascending and descending passes is technically feasible; however, the steep topography introduces several challenges. First, areas that are clearly visible from one orbit may be in shadow or appear foreshortened in the other, reducing data quality and spatial consistency. Second, since the topography is steep, the preferential direction of ground movement is often dictated by the slope of the terrain. Additionally, the 2D decomposition estimates vertical and east-west displacement components under the assumption that there is no north-south movement—an assumption that is frequently invalid in mountainous regions, where north-south displacement is commonly observed. Based on these issues, it was decided to use the orbits that yielded the best results for validation and mapping.* `

Are there other differences in the computation of the data in comparison to EGMS or other PSInSAR techniques ?

We added the following phrases: *`In general, there are technical differences in the computation of data across EGMS products, primarily due to the involvement of multiple groups in the project. Terrasigna's algorithms are more closely aligned with those used for the EGMS in southern Europe, which appear to offer better extraction of non-linear motion.* `

L. 139. Kinematics instead of dynamics.

We replaced *`dynamics`* with *`kinematics`*.

L. 140-141. Better formulate. The motion is measured along the SAR LOS direction, whereas the displacement of the rock glacier surface is expected to mainly occur along the slope or in the vertical direction, without that it is possible to evidence / discriminate them.

We added the following phrase: *`The motion was measured along the SAR LOS direction; however, the actual displacement of the rock glacier surface is expected to mainly occur along the slope or in the vertical direction.* `

One should also note that most rock glaciers in the area are north-south or south-north oriented, making that the satellite look angle is not well suitable for a fully consistent detection/analysis of the mass movement occurring along the slope and might conduct to underestimate of the actual displacements.

We added the following phrase: *`An important number of rock glaciers in the area are oriented north-south or south-north, which may lead to an underestimation of actual displacements due to the limited sensitivity of the satellite look angle to slope-parallel motion.* `

L. 147. "Due to the significant atmospheric noise in steep terrain…" or in areas with large elevation differences (whatever the steepness) ?

We added *`or in areas with large elevation differences`*.

Which mountain plateau ? Where ?

We deleted this.

L. 148. "… enabling PSInSAR measurements to cover the summits". What does it mean ?

We deleted this.

Figure 3 : is not legible for me. A zoom on the area of interest or just a part of it would be beneficial. Make also the legend legible. From there, it would also be very worth to describe the limitations and error sources of the applied technique in relation to the characteristics of the investigated terrain (slope aspect and steepness, vegetation, snow patches, …)

Based on suggestions from both reviewers, we modified fig.3 as follows: we zoomed in to the central area of Retezat mountains and we changed the background to a hillshade, for better visibility. We also enlarged the legend to be easier to read. We also outlined the associated limitations.

We added additional information on the PSInSAR technique, including its limitations in mountainous areas, at subchapter 3.2 (L187 - L204). Also, see comments above.

L. 158. "According to previous studies…" -> According to RGIK (2023) guidelines

We replaced `previous studies` with `RGIK (2023) guidelines`.

L.160. How were the MAs identified ? On the basis of Terrasigna's PSInSAR, I guess ?

Yes. We added `Terrasigna PSInSAR results`.

L.162-163. Again, refer to the RGIK (2023) guidelines instead of Barboux et al. 2014.

We replaced `Barboux et al., 2014` with `RGIK (2023) `.

L.163-164. I am a bit lost. To my understanding shadow and layover should prevent any (coherent) data acquisition. What are they doing here ? It would have been nice to make use of Figure 3 to illustrate the issue. Then, if you don't get valuable data, why to map the area as a moving one ? Not sure that this is following the RGIK recommendations.

We corrected this mistake by deleting this: `such as areas affected by shadows and layover`.

L. 165. Okay, but change the label of the velocity category to 0.3 – 1 cm/y (instead of < 1 cm/year).

We changed `<1` with `0.3-1` cm yr$^{-1}$.

L.166-167. I don't understand the meaning of the sentence "Because the number of … was low… we have (done the job)…in ArcGIS 10.8"

We replaced it with `The moving areas were manually digitized and compiled into an inventory using ArcGIS 10.8. `

L. 167-168. The ideas supporting the sentence are incorrect to me or difficult to understand. There is first an implicit association of permafrost occurrence and rock glacier creep, the latter being expressed by the surface movement (here measured in the SAR LOS direction only, without distinction between the along slope and vertical movement). It looks that one can expect that any piece of permanently frozen ground moves, whatever its spatial extent. Moreover, the lowermost size limit of mapping at 300 m2 (17 x 17 m),

that the authors applied, appears to be too small for them. But rock glacier creep is a process developing at about 20 m depth over significant areas. Moving areas of less than maybe about 1000 m2 (33 x 33 m) are very probably not consecutive to permafrost creep and should be disregarded in this purpose.

We added the following phrase: `Since many surface displacements in marginal periglacial regions result from active layer deformations (e.g., melt-induce subsidence), we have set the minimum area for an MA at 1000 m$^2$ to exclude those not associated to permafrost creep. ` All the moving areas smaller than 1000 m$^2$ were excluded and the statistical analyses were repeated.

L. 171. Validation of what ? Just write Differential GNSS measurements

We corrected according to your suggestion.

L.172 Provide for both rock glaciers the numbers used in Figure 1 and refer to the latter.

We added the numbers used in Figure 1.

It is extremely challenging to measure so low velocities (< 3 cm/year) with sufficient accuracy using dGNSS even on a 2-year basis. The 1 cm accuracy is not fully secured. It is usually expressing a standard deviation and concerns the horizontal (planimetric) positioning (the elevation is worse). The uncertainty of the velocity is larger, up to 2 cm/year (after one year). It is also extremely important to perform the measurement on stable (non-moving) points as well, not only on rock glaciers, in order to estimate this inaccuracy. Has this step been performed ?

Thank you for the comment. Yes, we are aware that measuring slow deformation processes is particularly challenging. However, we believe that dGNSS results still provide valuable information in such contexts. For reference, we also measured two control points located outside the rock glaciers, on stable bedrock.

We added the phrases: `Two points outside the boundaries of the rock glaciers, located on stable bedrock, were also measured to assess the horizontal accuracy of the DGNSS. The difference between the yearly measurements in both cases indicated an accuracy range of 0.3 to 0.6 mm/yr$^{-1}$. `

Another way of checking the quality of the data is to map the displacement vectors and to look for their consistency. Such kind of information must be provided. Otherwise, it is not possible to trust the data.

We have changed the symbols for the dGNSS markers to vectors that are proportional to the horizontal velocity and that also represent the direction of movement.

L. 182. The RGIK guidelines are not describing how to conduct the interpretation of interferograms, but they refer to another document.

We added a new reference (RGIKb, 2023) for the document describing how to conduct the interpretation of interferograms:

RGIKb: InSAR-based kinematic attribute in rock glacier inventories. Practical InSAR Guidelines (version 4.0., 31.05.2023), IPA Action Group Rock Glacier Inventories and Kinematics (RGIK), 33 pp, www.rgik.org, 2023b.

L.186. The BTS method does not allow to map the occurrence of permafrost with sufficient precision. The threshold defined by Haeberli >50 years ago were developed on glaciers/debris-covered glaciers/moraines (if I correctly remember) and were not suited for coarse blocky surfaces. There are many factors influencing the ground surface temperature under a well-developed snow cover, the first one being the constitution of the surface ground itself. A dry and porous bouldery surface (frequent on the investigated sites) tends to systematically provide lower BTS values. The early winter snow conditions are also extremely important. Air flow within the porous surface debris or deeper throughout the landform are likely to occur. So, the BTS method is very useful to evidence areas with colder ground surface temperature from those with warmer ones, and it must be mapped like that (not just using the -2/-3°C thresholds as on fig.12). The interpretation of the values themselves should then be done with great care.

We agree with you despite BTS method is still widely used to map permafrost distribution. We added the following phrases: *`However, in dry, porous bouldery surfaces where air convection and advection occur, this method lacks the precision needed to accurately map permafrost occurrence (Bernhard et al., 1998). Nevertheless, the BTS method remains highly effective for distinguishing areas with colder ground surface temperatures from those with warmer ones.*`

L. 202. How was WEqT computed ?

Note also earlier guidelines about WEqT by Schoeneich during the PERMANET project around 2012, which himself inspired from earlier works (e.g. PERMOS in Switzerland)

We added this phrase: *`WEqT refers to stable ground surface temperature period lasting at least two weeks, generally occurring in late winter, when snow cover exceeds 50 cm in thickness (Schoeneich, 2011).*`

L. 246. MAs are up to 10 cm/year according to Fig. 4. The authors would like to say the maximum PSI values are in the range of 5 cm/year (which is not around the uppermost detection capacity of the method ?) ?

We know that PSI can detect larger deformation; We wanted to say that 5cm/year is the maximum deformation measured using PSI and it was classified as between 3 and 10 cm/year. We wanted to use the established classification that is why we used the class 3 - 10 cm/year while the maximum values are at 5 cm/year

We changed L246 by adding a clarification. It now reads: "In the Retezat Mountains the MAs have velocities ranging from 0.3 to 5 cm yr$^{-1}$ (Fig. 4 and 5), which are classify into three velocity classes <1 cm yr$^{-1}$, 1 - 3 cm yr$^{-1}$, and 3 – 10 cm yr$^{-1}$."

What have been the rules to define the occurrence and the extent of a MA? And then to attribute a velocity class? This is unclear to me from Fig. 4.

According to RGIK (2023) guidelines, a moving area (MA) represents an area at the surface of the rock glacier characterized by relatively homogeneous velocity rates and consistent flow direction. We have set the minimum area for an MA at 1000 m$^2$. *`The velocity class characterizes the average yearly displacement rate recorded within a MA during the 2015-2021 period.*`

We have also modified Fig. 4 to be clearer.

L. 256. To what extent is the distribution of MAs among aspects related to the specificities of the investigated area (e.g. frequency distribution of aspects) ?

The frequency distribution of aspects in the investigated area is relatively well balanced with western aspects having a small dominance. This is poorly correlated with MAs which are predominantly having a N or NE aspect.

Figure 4: very difficult to read. Better to zoom on a smaller area

Based on suggestions from both reviewers, we modified Fig. 4 as follows: we zoomed in on the figure and we only presented two rock glaciers, one relict and one transitional. In order to be able to have the same color for corresponding velocity classes for the MA and PSI dots we choose to represent the MA with hashed polygons, instead of colored polygons.

Figure 5: the vertical scale in incorrect (cm); yellow curve is not legible; location map as well. It would be nice to have the location of the geophysical profiles on the location map.

We made the following changes to fig.5: we corrected the vertical scale to cm; we changed the color of the curve representing the 1-3 cm class from yellow to gold to make it more visible; we updated the location map by using the new RG contour and by adding the geophysical profiles to it; we also zoomed in on the location map in order to make it more visible.

I am wondering how the purple curve / time series has been produced. The winter jumps are about 2.5 to 3 cm, namely one phase on Sentinel C-band. How is the winter gap solved ? The behavior during the summer is also almost flat, at least in 2018-2019-2020. Is the total displacement really so large ?

We added the following caption to fig.5:

Displacement profiles over a period of 6 years (2015 – 2021) for 4 locations (identified in the location map with dots of corresponding color) representing the three moving areas and one for an area with no movement. The dots show the actual PSI displacement measurements, while trend lines (linear regressions) indicate long-term motion patterns. The displacement is measured relative to 2020. The downward trend can be interpreted as movement away from the sensor, which in our case represents a combination of vertical movement (i.e. subsistence) and horizontal downslope movement. The gap in point density along the trend line is due to the winter season, the measurements being performed in snow free conditions, usually from June to November.

Further explanation of the curves: Each dot represents the displacement between its corresponding date on the X axis and the reference point in 2020. For the magenta line the displacement is approximately 20cm (from 17 in 2015 to -3 in 2021) which over a period of 6 years averages to about 3.3 cm yr $^{-1}$. The trend line shows the direction of the movement (away from sensor) and it also shows that the movement is consistent over the study period. Some seasonal displacement trends can be observed (i.e. the displacement during the winter observed in the magenta line or the lack of displacement during some summer periods) but these trends are not consistent in the whole study area. Also tracking seasonal variations presents high uncertainty, thus we prefer to present only annual or multiannual displacement rates.

L. 269ff. The attribution of rock glaciers to the transitional category seems to have been overestimated. Looking at figure 10, I don't understand how have both big rock glaciers facing north close the center of

the map attributed to the transitional class. There is almost no movement on a very large part of their surface.

We have updated the rock glacier inventory. The biggest rock glacier (marked with 5 on fig.1) was reclassified as relict and the other rock glacier (marked with 7 on fig.1) was split into two rock glacier units (fig.1c), with the upper one being classified as transitional and the lower one being classified as relict.

Fig. 8. I would suggest to have the maximal elevation (2500 m a.s.l.) at the center, forming like a summit with slopes all around. Easier to read (at least for me).

We changed the figure to have the maximal elevation at the center.

Fig. 10 a and b) are both very difficult to read. They should focus on a smaller area or smaller ones.

We modified fig.10 to make it easier to understand. We zoomed in on the area, and we presented only one multi-unit rock glacier. We added explanations to both fig.10a and fig.10b to better point on the key areas that are to be observed in the figure. We added information to the legend, about the direction of the orbit.

Why the leftmost transitional rock glacier considered as such and different from fig. 9, where it is drawn as a relict one ?

It was a cartographical error in fig. 10 (it used a previous version of the RoGI). We updated the inventory used in all figures, including in fig. 10

Fig. 10a : what is this PSINSAR data, which looks to be different from those presented on fig. 4 ?

We changed the scale of fig.4 (it now presents a smaller area) and we adjusted the symbols in fig.10, thus the PSINSAR data in fig.4 and in fig.10a look similar.

Fig. 10b : "… the fringe cycle (bottom right) represents the change of colour". Please explain to the reader how to read the color (changes) according to the data which is used. What is the LOS (line-of-sight) ?

We added information to the legend, about the line of sight. We added the following information to the image caption: The fringe cycle (bottom right) represents the change of phase visualized by the colors. For ALOS-2 PALSAR-2 one phase cycle corresponds to a line-of-sight (LOS) displacement of 11.8 cm.

Note that the data of Fig. 10b is almost not used/discussed in the paper except two short sentences (L. 286 and 288). But the reader is left alone to read the data and calculate the displacement by him/herself.

It is important to highlight that we created and classified the MAs based on PSI with S1 scenes and we only used interferometry with ALOS-2 PALSAR-2 scenes to validate the results, especially their spatial distribution. However, we recognize that even if we are short on space we need to add some more information about the result presented in fig. 10b.

We added the following paragraph that further analyzes fig 10 and that compares PSI to InSAR: "If we compare the performance of PSI and InSAR we notice that, in an area with big variability in displacement, PSI can offer a more detailed mapping of the Mas, differentiating between relatively minor differences in their respective velocities. In Fig. 10b a MA is clearly identified and it was classified to <1 cm yr[1]. The same

area can be identified in Fig. 10a, where the majority of it is mapped as an MA classified as <1 cm yr[1], but we can also identify an MA classified as 1 – 3 cm yr[-1] and another one that is classified as 3 – 10 cm yr[-1]."

We also added another sentence to the fig.10 caption, stating the measured displacement for fig10b: "Fig. 10b presents the Galeşu RG where in the upper RGU we have one fringe over a five-year period (September 2014 to October 2019) accounting for a displacement of approximately 0.56 cm/yr. This is in line with the PSI measurements that have the biggest MA on the RG to be <1 cm/yr."

L. 298. Are the mean DGNSS (and not DGPS) velocities calculated over 2 years ? And how ? By computing the velocity between the initial and final positions (2-year span) – which should be the best way of doing – or by averaging the two annual velocities – not recommended ?

Are the velocity range and frequency for the rock glaciers only or do they also integrate points outside ?

We added this phrase: `The mean DGNSS velocities used in the analysis were calculated as the yearly displacement between the initial and final position over a two-year period. `We also measured two points outside the rock glaciers.

L. 303-4 and fig. 11a : there are to me several issues :

- GNSS : we would like to know the flow direction (in order to evaluate the coherence of the flow field and the data itself)

We changed the symbols for the dGNSS measurements to vectors that are classified using the same horizontal velocity classes as the MAs and that represent the direction of movement.

- At Judele, most GNSS points with velocities between 1 and 2.8 cm/y (this is mapped as 1 – 3 cm/y ) are located in either slower (< 1 cm/y) or faster (3-10 cm/y) PSINSAR-based moving areas (MAs). So, things are not working at all. Why ? Is the GNSS data failing ? The PSINSAR ? Both ?

We added this topic in the Discussion section (L501 - L507); however, a perfect match between PSInSAR and dGNSS data is not possible due to the different monitoring intervals (2015–2021 for PSInSAR and 2019–2021 for dGNSS).

- The MAs mapping appears weird. Purple areas (3-10 cm/y) are surrounded by green ones (< 1 cm/y), the intermediate yellow ones (1-3 cm/y) being missing. The latter are however located outside of the green ones. Such a flow field is not consistent.

Thanks for the comment. We updated the RoGI and the MAs. In the case of Judele rock glacier one MA was erroneously classified in the velocity category. The new mapping of MAs provides a more consistent movement pattern but it still remains the rock glacier with the most heterogeneous MAs in the Retezat Mountain. This might be due to its orientation being a source of uncertainty (see the paragraph below) but we needed to be consistent with the methodology used on the other RGs in this study.

- One should also note that the rock glacier could be expected to move to the NE (maybe partly to the N), the use of InSAR data in descending mode in not adequate.

We acknowledge that the descending mode has reduced sensitivity to eastward motion, which is relevant given the NE-oriented displacement of the rock glacier. However, due to complementary validation with

dGNSS data, the descending InSAR data were considered sufficient for the scope of this study. Future analyses will aim to integrate ascending track data to improve sensitivity to the displacement direction.

L. 309. See my comment in the related methodological section

We have revised this section according to your suggestions and now present BTS values in terms of cold and warm surface areas, without referencing permafrost probability thresholds.

L. 310. "highest" means "warmest", right ?

We replaced `highest` with `warmest`.

Fig. 12. Provide actual BTS values in order to see where the colder/warmer areas are (as commented earlier). I am also sceptic about the MAs mapping (it appears to be too detailed considering the uncertainties of the method)

We modified fig.12 to have the actual BTS values (in increments of 1° C). We have also reconsidered MAs, using the recommended minimum surface of 1000 m².

L. 325. "in all the years" instead of "in all the seasons" ?

We replaced `seasons` with `years`.

Fig. 13a: better to provide MAGST under the form of running means instead of just single values (hydrological year)

Thank you for your suggestion. We represented MAGST as a running mean.

Fig. 13c: Great dataset ! There is probably more to exploit from it. For instance, the zero curtain phase (snow melt phase) appears to be pretty long, confirming the installation of thick insulating winter snow cover. Despite it, in many years and on many sites, there are still temperature variations during most of the winter. As in 2013/14, they even appear to be inverted for a part of the winter between Pietrele and Judele. It would be also nice (valuable) to compare them to those of the air temperature (if possible). What are these temperature variations telling us ?

Thank you for your comment. We have expanded the section dedicated to this analysis and included a comparison between GST and air temperature. In some cases, we observed inverse thermal relationships, supporting the presence of advective heat fluxes within the coarse debris during winter.

L. 339. How is it possible to get consistent information on this 3-4 m superficial layer from an ERT profile using an electrode spacing of 4 m ?

It is true that a 4 m sensor spacing restricts the overall resolution capacity of both methods and increases the uncertainty for correct identification of the thickness of very shallow layers. We adjusted the text accordingly and now mention more generally 'an up to 5 m thick surface layer' to avoid overinterpretation.

L.341. Could this second patchy resistive layer (please refer explicitly to the figure and show it) alternatively be porous debris ? Why is it talked about "remnants" of ice-rich permafrost and not just about patches ? Remnants from what ?

Thanks for pointing to this unclear statement. In fact, the layer of interest was originally highlighted by dashed circles, but we realize now, that this was almost invisible. We adjusted the color map (following suggestions of reviewer 1) and improved the overall labelling of relevant zones in Fig. 14.

Regarding the potential for porous debris: yes, this is of course a probable interpretation. With respect to the very coarse-blocky surface of the rock glacier, which leads to a high porosity in this surface layer, and from a geomorphological point of view, a transition to an even more porous layer underneath seems however less probable than a transition into a layer with similar porosity but higher ice content. With remnants of ice, we mean remnants of an originally thick ice-rich layer, which already experienced substantial degradation. We changed the wording to avoid confusions.

L. 342. This layer should correspond to the rock glacier body, not to the bedrock (what is the resistivity of the bedrock ?). Note that with 20-30 kohm.m the frozen state of the medium cannot be excluded.

Thank you for this comment. We agree that values around 20-30 kohm.m do not contradict a potentially frozen state of the rock glacier body. However, the seismic data indicate high velocities in the same zone, which – in combination with the ERT result rather point to bedrock than frozen porous debris. We noticed, however, that this was not sufficiently explained in the text and improved the argumentation accordingly.

L. 346. One should note that talus (as relict and some transitional rock glaciers) can be ventilated and by this way, deeply frozen (unsaturated). I don't know if it makes sense regarding the investigated site.

Thanks for the comment. Yes, this is actually possible, but not the focus of this study. We now mention this possibility in the text.

Fig. 14a. It looks that the "ice-rich ?" and the corresponding arrows are shifted. They are some almost invisible dotted lines (same on fig. 14b), but we don't know what it is.

See our comment further above. We improved the overall labelling of the plot.

It would be nice to make the color scale more accessible for the reader (e.g. what is the yellow actually representing). There are 3 orders of magnitude represented and this is almost impossible for the reader to relate the colors with an actual resistivity value. Maybe some contour lines could be inserted. Same for Fig. 14c and 14d.

Thank you for the proposition. With the applied triangular mesh of the PyGimili framework contour lines are unfortunately not easily applicable, as they would align along the triangular shape of the mesh cells. We therefore prefer not to add contour lines here, but we improved the appearance of the color bar.

Fig. 14d. From the color scale, it is impossible to read this data (we don't know what values are mapped). We don't know the unit as well.

Thanks for spotting this error. We added the unit (%) to the color bar and also refer to the min/max values indicated in the panel in the figure caption. Further, we adjusted the color map to show discrete intervals of 5 %, which hopefully improves the readability of this panel.

I am also very surprised that the "bedrock" shoulder (low resistivity/high velocity) provides as much ice content as the rock glacier (namely the debris-constituted ground).

Yes, the modelled ice content distribution does not exactly match the expectations of the individual interpretation, for several reasons: a) the ERT data have a very limited quality due to the extremely coarse-blocky surface and dry measurement conditions, b) the resolution capacity of both ERT and RST is limited because of the 4 m sensor spacing, making it difficult to resolve e.g. thin layers, and most importantly c) the drawback of the rock-ice ambiguity of the petrophysical formulation of the PJI. This rock-ice ambiguity is due to the fact, that Archie's law (used in the current formulation of the PJI) relates the measured resistivities only to porosity, the resistivity of the pore water and the saturation of the pore space, whereas the resistivity of ice and rock are not part of the equation and are only constrained by the equation used for the seismic velocity (see Hauck et al. 2011 or Mollaret et al. 2020 for details). Even if this was not a focus of this paper, we see that it is still necessary to mention this problem and we added a more detailed explanation for better understanding.

L. 357. As commented above, 20-30 kohm.m is far from excluding frozen ground to occur.

We agree and reformulated the interpretation accordingly, in order to mention both possible interpretations.

L. 359. So, no ice-rich permafrost occurrence.

Yes, as mentioned earlier, we added more detailed explanations in the text, that the resolution capacity would not be sufficient to resolve thin ice-rich layers or anomalies, but that the overall low ice content rather point to low bulk ice contents, which do not exclude thin layers with saturated conditions. The resulting low ice contents are actually mainly confirming the absence of a massive ice core and thus indicative of a strongly degraded situation in contrast to an active rock glacier. We changed the labels in the figure from 'ice-rich?' to 'ice?' to avoid confusion.

Discussion

L. 370. A very patchy motion signal might either be related to noise (data uncertainty) and superficial/shallow movement, but not to rock glacier creep (deformation at a shear horizon at about 20 m depth). The a priori association of (INSAR-detected) surface movement to permafrost occurrence must also be avoided.

We deleted this phrase. Based on the geophysical results at the Galeșu rock glacier, we suggest that surface displacements at this site are likely caused by processes other than permafrost creep (subsidence, solifluction, blocks sliding). However, the conditions during the geophysical survey were not ideal, and the results should be interpreted with caution. While we cannot entirely rule out the presence of permafrost creep without additional geophysical data from other rock glaciers and more precise dGNSS measurements, we believe that the consistent downslope movement of dGNSS markers through the rock glacier fronts at Judele and Berbecilor supports the interpretation of permafrost creep. In contrast, processes such as subsidence or block sliding typically result in more irregular or chaotic displacement patterns.

L. 380. Matching of DGNSS and PSInSAR cannot be supported (see comment Fig 10).

We agree. We referred to this in the subsection 5.1.

L. 383. The velocity is probably too small for small but significant interannual variations to be detected.

Thanks for the comment. We decided to eliminate this figure and the corresponding analysis.

L. 384-6 and Fig. 15 : There is a big issue to me. There are just two "families" of points : those moving around 0.5 cm/y and those (the majority) moving around 3 cm/y. And nothing between. It does not make sense. There should be intermediate values. The range is close to the SAR phase (half a wave length, about 2.75 cm). I suspect an issue regarding the solving of this phase ambiguity. We would like to better know how these values have been computed and where the points are located on the rock glacier.

You are right. Fig. 15 was deleted.

L. 400. What is precisely the rule which has been applied to categorize a rock glacier as transitional or relict ? Would it not be also adequate to separate between rock glacier units and systems are recommended by the RGIK guidelines ?

We have followed the definition of RGIK (2013), but unfortunately this is a bit ambiguous and open to interpretation. `*Transitional: rock glacier with slow movement only detectable by measurements or movement restricted to areas of non-dominant extent…If adequate kinematic data is available: a transitional rock glacier shows little to no downslope movement over most of its surface. Relict: rock glacier with neither geomorphological evidence nor detection of current movement associated with permafrost creep …If adequate kinematic data is available: a relict rock glacier shows no detectable downslope movement over most of its surface, and the geomorphological characteristics are as described above. *`(RGIK, 2023, p.10).

The definition does not strictly pertain to the surface extent of the moving areas within the rock glaciers, but rather in both cases, it emphasizes that these areas should not have a dominant extent. We classified as transitional only those rock glaciers that exhibited movement within the 1–3 cm/yr and 3–10 cm/yr velocity classes, in order to ensure that no noise was introduced into the classification. Where applicable, we also separated rock glaciers into distinct units to better reflect the spatial variability of movement (ex: Galesu or Valea Rea).

L. 404. What is the sense of this comparison with Brencher et al. ?

We deleted this phrase.

L. 406. The occurrence of the 5-10 m depth patches of ice-rich permafrost is not supported by the RST data.

Yes. As we adjusted the overall interpretation as requested, we also adjusted this part accordingly.

L. 414ff. GST and BTS suggest (and not confirmed) the occurrence of permafrost (see my comments above).

In addition, the current MAGST are roughly about -0.5°C. Taking into account a presumed climate warming by +1.5°C over the last decades, they could have been around -2°C before. It means that extended permafrost occurrences in the rock glaciers can be expected and that deep permafrost (to say down to 50 m or so) cannot be excluded. The absence of ice-rich (ice supersaturated) permafrost conditions at depth (in the investigated rock glacier) does not mean the absence of (ice non-saturated) permafrost, which is highly susceptible to occur according to the geophysical results and the GST values. A more detailed analysis of the BTS mapping and the GST behavior in wintertime might provide some clues about possible

convective and/or advective air flow processes within the rock glacier landform, which could contribute to explain why the rock glaciers are so cold.

We added the following text: `Considering the current MAGST of approximately -0.5 ºC and assuming a climatic warming of about +1.5 ºC since the late 19th century (Allen et al., 2018), it is likely that these rock glaciers had a MAGST around -2 ºC during the pre-industrial period. At such low temperatures, widespread permafrost conditions would have been expected, and the presence of deep permafrost cannot be ruled out. However, accelerated warming in recent decades has resulted in permafrost warming, particularly in ice-poor bedrock, at rates comparable to the increase in air temperature (Noetzli et al., 2024). BTS measurements and GST patterns observed during winter suggest ongoing convective and advective air flow processes that maintain cold ground conditions and support the persistence of ice non-saturated permafrost in the Retezat Mountains. `

L. 456. Conclusions are fine and sound. It might be that they could be slightly altered after revision of the paper. The authors must insure that the abstract is strictly based on the paper's conclusions.

We have updated the conclusions after revising the paper.

References mentioned in my review (not already cited in the manuscript):

Cicoira et al. (2020). A general theory of rock glacier creep based on in-situ and remote sensing observations. https://doi.org/10.1002/ppp.2090

Delaloye et al. (2013). Rapidly moving rock glaciers in Mattertal. In: *Graf, C.* (ed.) Mattertal – ein Tal in Bewegung. Publikation zur Jahrestagung der Schweizerischen Geomorphologischen Gesellschaft 29. Juni – 1. Juli 2011, St. Niklaus. Birmensdorf, Eidg. Forschungsanstalt WSL. 21-30.

Eriksen et al. (2018). Recent acceleration of a rock glacier complex, Adjet, Norway, documented by 62 years of remote sensing observations. Geophys. Res. Lett., 45, 8314–8323

Hartl et al. (2023). Multi-sensor monitoring and data integration reveal cyclical destabilization of the Äußeres Hochebenkar rock glacier. https://doi.org/10.5194/esurf-11-117-2023

Hu et al. (2025). Rock Glacier Velocity: An Essential Climate Variable Quantity for Permafrost. https://doi.org/10.1029/2024RG000847

Kääb and Røste (2024). Rock glaciers across the United States predominantly accelerate coincident with rise in air temperatures. https://doi.org/10.1038/s41467-024-52093-z

Noetzli et al. (2024). Enhanced warming of European mountain permafrost in the early 21st century. https://doi.org/10.1038/s41467-024-54831-9

Pellet et al. (2024). Rock Glacier Velocity. In "State of the Climate in 2023". Bull. Am. Meteor. Soc. 105, 42-44. https://doi.org/10.1175/2024BAMSStateoftheClimate.1

Roer et al. (2008). Observations and considerations on collapsing active rockglaciers in the Alps, Proceedings of the Ninth International Conference on Permafrost, July 2008, Fairbanks, Alaska, 2, 1505-1510.

Schoeneich (2011). Guidelines for monitoring GST - Ground surface temperature. PERMANET. https://www.permanet-alpinespace.eu/archive/pdf/GST.pdf

Thank you. We have considered all these recommendations.

---

## Author Response (AR2)

**RC1: 'Comment on egusphere-2024-3262', Anonymous Referee #1, 27 Jun 2025**

Author comments (AC) in blue

I appreciate the additional explanations provided by the authors. The contents of the manuscript are now easier to follow and the discussion and interpretations are more focused and clear. The figures are much improved, although I still have some quibbles (see below). My remaining comments are minor and mostly about technicalities related to the figures, with a few additional questions. I believe this work is a valuable overview of regional rock glacier state and will make a nice contribution to ESurf.

Thank you very much for your encouraging and constructive comments. We appreciate your feedback and have carefully considered and addressed all your suggestions in the revised manuscript. We sincerely hope that our responses to each of your points are satisfactory and that the revised version meets the standards for publication.

L59 increased rock glacier velocities has been linked to warmer climate

—> have been

Thank you. The typographical error has been fixed in the revised version.

L66 consider adding a sentence to explain how destabilization differs from "normal" permafrost creep to indicate why such high velocities are reached.

We added the following phrase: "Destabilization typically involves a sudden, pronounced acceleration of a section of the rock glacier, with displacement rates increasing by up to two orders of magnitude and surface manifestations such as cracks, scarps, and crevasses reflecting enhanced internal strain between stable and unstable areas (Marcer et al., 2021; Hartl et al., 2023)."

L88 The sentence on changing temperatures might fit better in the following section where you discuss the climatology of the study region. Suggestion: Move this sentence to line 103 in Section Study Area.

We moved this phrase to Study Area.

L 90, 94: I suggest using "kinematics" instead of "dynamics" here and throughout the manuscript.

Dynamics typically refers to underlying processes and causes (why is it moving), while kinematics describes the effect (how is it moving). You mostly describe the latter.

It has been corrected in the revised manuscript!

Fig. 1: Thank you for adjusting this figure. I still have a few comments/suggestions:

In panels a and c the text size in the legend is still really small. Can this be made larger? In Panel b, the dark blue color of the permafrost extent does not seem to match the color in the legend - it would be easier to understand if the same color is used. The hashed circles for GST and GNSS measurements are difficult to visually differentiate at site 8. More contrasting colours might help. What is the difference between the thicker and thinner black outlines? Consider adding an explanation to the caption or legend.

In panel c you have added "extended" and "restricted" outlines - it is not clear to me where these come from and what the difference is. Is this explained somewhere? Consider explaining the meaning of "rock glacier units". I believe that site 7 was presented as one landform in the previous version whereas it consists of 2 units in the revised manuscript.

We adjusted Figure 1 as follows:

In panel (a) we deleted one item (Southern Carpathians contour) from the legend and made the rest bigger and easier to read

In panel (b) we corrected the colour, for modelled permafrost extent, in the legend to match the colour in the map; We deleted the four symbols for the ground-based measurements that were performed on different rock glaciers (this information can be found on a new table added to the text).

In panel (c) we zoomed in even more and also, we made the text in the legend bigger and easier to read

We adjusted the caption accordingly.

We added to the caption the explanation for the thicker lines in the inventory.

We introduced the "extended" and "restricted" outlines at the recommendation of Reviewer2 in order to comply with the recommendation from the RGIK (rock glaciers inventory and kinematics) action group. The extended outline is the former outline, while the restricted outline follows the top of the front and margins of the rock glacier. The rock glacier unit was introduced, again, to follow the recommendations of RGIK.

We made small updates to the rock glacier inventory to comply with the recommendations from the RGIK. This include mapping rock glacier units and multi-unit rock glaciers (this being the reason why we also introduced the term "rock glacier unit". We have split the Galeşu rock glacier (no 7 on fig. 1) into two units based on kinematics and morphology.

L136 check formatting of citation

We revised all the citations and made corrections were needed.

Fig. 3: The colours and overall legibility of the figure are much improved, thank you. The text in the legends and grid are still relatively small but legible when the pdf is displayed at full size.

For Figure 3 we changed the layout from landscape to portrait, with (a) above (b) instead of side by side. This way we had space to increase the size of the legend, including text, and of the scale bar text

L217: The next step was to assign velocity classes to moving areas considering the standardised velocity classes

Suggestion to rephrase: —> The next step was to assign standardised velocity classes to moving areas

Thank you for the suggestion. It has been corrected in the revised manuscript!

L219 -222: check font and text size

Thank you for pointing out this oversight. This mistake has been corrected in the revised manuscript!

L 222: PSInSAR-based surface displacement of ≤ 0.3 cm yr-1 were assigned to the "no movement" category, …

This reads like a contradiction to the sentence two lines up that lists the categories. Are "undefined" and "no movement" the same categories or do you differentiate this somehow? Maybe add "no movement" to the list of categories.

Thank you for the suggestion. The 'Undefined' category was assigned exclusively to moving areas (MAs) exhibiting inhomogeneous velocity patterns, as explicitly stated in the Methodology section (two sentences further). In contrast, the 'No movement' category refers to areas without detectable displacement and therefore cannot be considered a type of moving area.

Fig 10: text size could be increased. The annotations in 10b would be easier to see with a backdrop as in 10a. The small text in the "fringe cycle" inset in 10b is very hard to read.

We increased the font size for the legend for fig. 10 (a). We modified the graphics for the "fringe cycle" and we increased the font size for all the text in the legend for fig. 10 (b). We added a white background to the text in the main window of fig. 10 (b) to increase its visibility.

Fig 11: as above, I am wondering what the difference between the "extended" and "restricted" outlines is.

We introduced the "extended" and "restricted" outlines at the recommendation of Reviewer2 in order to comply with the recommendation from the RGIK guidelines. The extended outline is the former outline, while the restricted outline follows the top of the front and margins of the rock glacier.

L390 while only three falls within —> fall within

Thank you.  The typographical error has been fixed in the revised version!

Fig 12: label size for panel a and d is small, scale bar text is small

We increased the font size for labels (a) and (d). We also added a white background for labels (a), (b), (c) and (d) for better visibility. We increased the font size for the scale bar text on panels (a), (b), (c) and (d).

L521 and following: statements such as "very cold" and "considerably warmer" could be quantified by giving corresponding measured values.

Thank you for the suggestion. We added corresponding measured values.

**RC1: 'Comment on egusphere-2024-3262', Anonymous Referee #1, 27 Jun 2025**

 **Author comments (AC) in blue**

I have very much appreciated the thorough consideration of the reviewers' comments and the efforts made the authors for improving their manuscript. To me, the paper has well gained in both consistency and quality. It is now in a very good shape.

I have very few final comments on the text and would still suggest further improvement of some figures (see below).

Otherwise, all is fine. Very good job !

Thank you for your valuable and constructive comments. We have carefully considered all your remarks and addressed each of them in the revised version of the manuscript. We hope that our responses are satisfactory and that the revised submission will meet your approval for publication.

l.32 – « At this site, the recorded surface displacements are more likely the result of ice-melt-induced subsidence….” If this is an ice-poor site, ice-melt-induced subsidence cannot really occur. In fact, what is actually causing the motion observed at the surface it unknown. I agree to assume that is not related to permafrost creep. So, the rock glacier cannot be classified as transitional, but only as relict or relict uncertain.

Thank you. This phrase has been corrected in the revised manuscript.

l. 33 – "…, solifluction, or the tilting and sliding of blocks within the active layer". I would suggest to simplify and just write that you presume, because of the heterogeneity of the observed surface displacement, that the motion is primarily occurring by deformation at shallow depth, namely in the active layer.

This part of the phrase has been deleted.

l.34 – at two other rock glaciers (add "other")

This phrase has been corrected.

l.54 – Ice-melt subsidence cannot occur in the active layer, but only at the permafrost table if the active layer is gaining in thickness.

This phrase has been corrected.

l.62-63 – "According to Necsoiu et al. (2016), slow-moving rock glaciers in the Southern Carpathians exhibited increased velocities between 2007 and 2014, attributed to rising permafrost temperatures". Interesting and important but not placed at the right location. This is here a paragraph about rock glacier creep in general. The sentence should come 2-3 paragraphs later or in Section 2, when describing the regional settings. Provide also numbers.... e.g. Slow-moving RGs in the S. Carp. displace up to x cm/year. They exhibited a velocity rise of x % between 2007 and 2014...

This phrase has been moved to the Study area as you suggested.

l.77-78 - No movement means a kinematically relict rock glacier according to RGIK, not a transitional one.

Thank you for the suggestion. It has been corrected in the revised manuscript!

l.88-89 – "Above 2000 m in the Southern Carpathians, the 1991-2020 climatological period was 0.8 ºC warmer than the 1961-1990 baseline (Berzescu et al., 2025)." Should be placed in section 2.

It has been moved in section 2.

l.91 – « more broadly » or more precisely ?

Thank you for the suggestion. `More broadly` has been replaced with `more precisely`!

l. 115 – Fig. 1 - The color do not fit with the map for the modelled permafrost extent. Those in the dots are impossible to read and distinguish. Maybe add a table instead of using this color code.

We have to zoom a lot to make the text in maps (a) and (c) readable.

We adjusted fig.1 as follows:

In panel (a) we deleted one item (Southern Carpathians contour) from the legend and made the rest bigger and easier to read

In panel (b) we corrected the colour, for modelled permafrost extent, in the legend to match the colour in the map; We deleted the four symbols for the ground-based measurements that were performed on different rock glaciers (this information can be found on a new table added to the text).

In panel (c) we zoomed in even more and also, we made the text in the legend bigger and easier to read

We adjusted the caption accordingly.

l. 119 – "The rock glaciers that are discussed in the present paper are numbered (1 - Stânișoara, 2 - Bucura, 3 - Pietrele, 4 - Pietricelele, 5 - Valea Rea, 6 - Păpușa, 7 - Galeșu, 8 - Judele, 9 - Berbecilor), and the ground based measurements that have been performed on

each of them are represented by a composite symbol near the number". Not necessary if an adjacent table is used.

We removed the symbols from fig.1b and we introduced Table 1, which provides information on the investigated rock glaciers, their activity status and the type of measurements conducted on them.

l. 131 – Fig. 2 - Nice views, but an outline should be provided for the rock glaciers (don't leave the reader to interpret the landscape by him/-herself only).

Which rock glaciers are relict ? transitional ?

Thank you for the suggestion. We have added which rock glaciers are relict / transitional.

l. 132 – It might be useful to use the same numbering as in Fig. 1, e.g. 3 - Pietrele..

Thank you for the suggestion. Similar numbering has been added.

l. 156-157 - Remove the URL and add the reference in the bibliography.

We revised all the citations, including these one, and made corrections where needed.

l. 188-191 – "In general, there are technical differences in the computation of data across EGMS products, primarily due to the involvement of multiple groups in the project. Terrasigna's algorithms are more closely aligned with those used for the EGMS in southern Europe, which appear to offer better extraction of non-linear motion". Could be removed or placed one paragraph earlier.

This phrase has been removed.

l. 192 – 181 images in both modes (ascending and descending) ?

In the fourth paragraph in the 3.2 section, we state that the EGMS product (fig.3 a) uses both ascending and descending orbits while the TerrasignaPSInSAR (fig. 3b) uses only Path 80 descending of S1.

l. 200-202 – "Due to significant atmospheric noise in steep terrain or in areas with large elevation differences, a reference point at a similar elevation to the rock glaciers was chosen on the mountain summit. This approach minimizes atmospheric phase differences, improving

coherence and reliability of the PSInSAR measurements". Already written a few paragraphs before.

This phrase has been removed.

l. 203 – "underestimation of actual displacements" : underestimation of the along-slope displacement, but not in case of subsidence, and without any option to distinguish them.

Thank you for the suggestion. It has been corrected in the revised manuscript!

l. 205 – Fig. 3. – All is very small (text, legends), as well as the PSInSAR dots. Maybe zoom to a smaller area and make the dots larger.

For Figure 3 we changed the layout from landscape to portrait, with (a) above (b) instead of side by side. This way we had space to increase the size of the legend, including text, and of the scale bar text.

b) This is just in descending mode, right ?

The TerrasignaPSInSAR is only presented for descending mode. We stated this in the 3.2 section, paragraph 4  (also, see comment above).

l. 216 – about delination, just note that the RGIK guidelines for MAs recommend that, when MAs with various velocity classes co-exist, faster MAs are always embedded into slower MAs. This recommendation has not been applied here.

Thank you for your suggestion. We highly value the RGIK guidelines and fully acknowledge their importance in reducing discrepancies across rock glacier inventories worldwide. These standards are essential for fostering consistency, particularly in regions with well-developed and active permafrost dynamics. However, as the guidelines themselves acknowledge, applying uniform criteria in marginal periglacial environments—where displacements are minimal and often near the detection threshold—presents significant challenges. In such contexts, methodological flexibility becomes necessary. As is often the case in science, it is inherently more complex to study phenomena at the edge of their environmental or process-driven limits. This also explains the relatively limited number of studies addressing the behavior of rock glaciers in these transitional zones.

As mentioned in our previous response, even within RGIK, ambiguities remain—such as in the definition of transitional rock glaciers, which includes those with "little to no downslope movement over most of their surface." The guidelines themselves acknowledge this

complexity by stating that "subjectivity must be acknowledged as part of the rock glaciers mapping process." We consider this recognition fundamental in the context of our study.

Regarding the delimitation of moving areas (MAs), while we could not identify the specific reference you mentioned in Section 6.2.1 of the RGIK guidelines (Moving areas chapter), we acknowledge that other sections may contain broader interpretation. However, in our case, the velocity fields are highly uniform (ranging from 0.3 cm/year to 5 cm/year), which minimizes concerns regarding heterogeneity. The observation you raised becomes more relevant in studies aiming to compute velocity for entire landforms, especially when internal velocity contrasts are significant. In contrast, our study's objective was not to assign a representative velocity to each rock glacier, but to delineate MAs and use this information to classify landforms as relict or transitional. Concerning the delineation of individual MAs, we refer to the RGIK guidelines, which state: "The minimum extent of a MA depends on the spatial resolution of the data input, but also on the size of the considered landforms. It is based on the operator's judgment." In line with previous studies (e.g., Bertonne et al., 2012), we argue that mapping individual MAs provides valuable insights—particularly in cases where localized destabilization occurs. For example, if a clearly defined section of a rock glacier is accelerating (e.g., moving at several meters per year), while the rest is creeping at decimetric rates, delineating this area separately is scientifically justified and important for future monitoring. Merging such a feature into a broader, slower-moving MA would obscure a significant dynamic signal.

While we fully support the use of RGIK as a guiding framework, we believe our approach remains consistent with its flexible principles and is appropriate for the specific conditions of our study area and objectives.

l. 217 – standard instead of standardized

Thank you for pointing out this oversight. This phrase has been corrected.

l. 238-239 – Simplify the sentence as follows : The velocities used in the analysis were calculated between the initial and final position over a two-year period.

Thank you for the suggestion. It has been corrected in the revised manuscript!

l. 244-246 – "For the interpretation of the interferograms, we followed the practical guidelines of the IPA Action Group Rock glacier inventories and kinematics (RGIK, 2023b) ». It might be worth of shortly describing what it means. Most readers surely don't know.

We changed this sentence with the following text: "We masked areas affected by geometrical distortions and we used only interferograms from snow-free periods. For the interpretation

of the interferograms we only assign kinematic attributes when signals were reliable (e.g. on descending mode), we identified and delineated moving areas independently and then we linked the MA to the existing rock glacier units (RGIK, 2023b)."

l. 313-314 – "and were classified into three velocity classes: 0.3 – 1 cm yr-1, 1 – 3 cm yr-1, and 3 – 10 cm yr-1". Repetition of what already written in lines 218-219.

Thank you for the suggestion. It has been corrected in the revised manuscript!

l. 315 – remained "consistent" or constant ?

Thank you for the suggestion. It has been corrected in the revised manuscript!

l. 315 – about the "trendlines in Fig. 5". My previous answer (1st revision round) has not been answered (or I missed or misunderstood the explanation) : if we cannot trust the seasonal variations, why could we trust the winter ones (long data gaps), on which the trendlines are mostly based on ?

Thank you for the question. The trendlines in Figure 5 are based exclusively on snow-off periods, i.e., periods with valid displacement measurements. While interannual variability in rates is evident, the overall trend is derived from seven distinct snow-off periods.

l.337 – Fig. 4 – Outside of the rock glaciers, there are various noisy areas with values balancing between < 0.3 and 1-3 cm/yr. Why to consider classes < 0.3 cm/y and 0.3 to 1 cm/y as consistent or to limit them at the rock glacier drawn boundaries ?

Of course, there are moving areas outside of the rock glaciers, but for the purpose of the present study, we only present the moving areas from inside the rock glaciers. This decision was made to simplify the presented data in order to make the figures easier to read and understand.

PSI velocity : the legend must be adapted to the main text (< 0.3 cm/y , 0.3 to 1 cm/y…)

We adapted the legend on all figures to have the same MA categories as the main text.

l. 340 – Fig. 5 – Which rock glacier is represented ? Make the legend of the inserted map larger (and readable).

"PSI velocity class" (what is this ?) or MA velocity class ?

We modified fig. 5 as follows:

- We removed the geophysical profiles from the inserted map and from its legend and we increased the font size for the legend in order for it to be easier to read.
- We changed the representation of moving areas, from hushed polygons to hollow polygons, in order to make the dots for the location of the profiles more visible.
- We changed the legend title for the main graph from "PSI velocity class" to "MA velocity class"

l. 341-342 – "4 locations (identified in the location map with dots of corresponding colour" - I don't really see them, at least the yellow and green ones, which also are difficult to read on the chart.

We changed the representation of moving areas, from hushed polygons to hollow polygons, in order to make the dots for the location of the profiles more visible. (see paragraph above)

l. 345 – same question as in line 315 about the "gap" : how is the gap solved ? Because it looks that it is especially these gaps which define the trendlines.

Same answer as above:  the trendlines in Figure 5 are based exclusively on snow-off periods, i.e., periods with valid displacement measurements. While interannual variability in rates is evident, the overall trend is derived from seven distinct snow-off periods.

l. 364 – What are "areas with high variability in displacement" ? Spatial or temporal variability?

Thank you for the suggestion. Spatial has been added!

l. 365 – "more detailed mapping of the monitoring areas (MAs), allowing for the differentiation of relatively minor velocity differences". MAs means "moving areas" and not "monitoring areas".

Thank you for the suggestion. It has been corrected in the revised manuscript!

l. 372 – Fig. 10 – Provide both mode (Asc. / desc ?) and time of the represented interferogram. Make texts and legends larger and/or better readable.

Thank you for the suggestion. We increased the font size for the legend for fig. 10 (a). We modified the graphics for the "fringe cycle" and we increased the font size for all the text in

the legend for fig. 10 (b). We added a white background to the text in the main window of fig. 10 (b) to increase its visibility.

We stated in the main text, at Line 173-175, that we computed interferograms for both modes, ascending and descending, but we only present the descending mode, since it is the one that produced the most reliable results. Unfortunately, ascending paths proved too noisy over the Carpathians, and as stated in the Methodology section, we chose to rely exclusively on descending paths for mapping the moving areas. We also added this information in the figure caption. The time of the interferogram has been added after the first review.

l. 376 – What is a RGU ?

We modified the caption, we replaced "RGU" with "rock glacier unit".

We added an introduction to the RGU abbreviation at Line 273-274, RGU – rock glacier unit. We introduced the term after the first review in order to align our inventory to the RGIK recommendations.

l. 377 – "0.56 cm/y" - Where does this value come from ? There is about half a fringe (light blue to pink-violet), meaning about 5-6 cm in 5 years or 1-1.2 cm/yr.

We deleted the information on velocity from the Fig. 10 caption, since the objective of the figure is to compare the distribution of the moving areas between the two methods, PSInSAR on S1 and InSAR on ALOS-PALSAR2 .

l. 378 – largest instead of "biggest". According to my previous comment, is this PSI-based MA velocity class still in line with the present observation ?

Thank you for the suggestion. It has been corrected in the revised manuscript!

We decided to remove the information about the velocity derived from InSAR on ALOS PALSAR.

Revising the velocity from the interferogram in fig10b to between 1 to 1.2 cm/year it's still comparable with the velocity presented in fig10a, where there are 3 moving areas (0.3-1cm, 1-3cm, 3-10cm), when we consider the small velocities and the possible error rates for ALOS PALSAR (0.2 – 0.5 cm/year).

This is also another argument to only present the comparison for the spatial distribution of moving areas and to simplify by eliminating the comparison for velocity classes

l. 416 - +0.8°C is not negative. Maybe slightly adapt the sentence.

Thank you for the suggestion. This phrase has been adapted!

l. 455 – The ice content in d) appears to be in contradiction with b) and c), as described in the main text. There is almost no ice where ice is supposed to occur. Does it still make sense to keep the "ice ?" mentions on b) and c) ?

Yes, we agree that the labels are confusing and removed them from b) and c) accordingly.

l. 512 – "Unlike other regions (e.g., Central Italian Alps, Eastern European Alps, Himalaya) where there is a considerable elevation difference between active/intact and relict rock glaciers (Kellerer-Pirklbauer et al., 2012; Scotti et al., 2013), the Retezat Mountains exhibit a significantly smaller separation". But there is no active rock glacier at Retezat, just some transitional or relict uncertain ones on one side, and relict ones on the other. So, the sentence does not really make sense.

Thank you for the suggestion. This phrase has been deleted!

l. 533 – "…but likely suggests negligible ice content". Maybe better to write "… but likely suggest an insufficient ice content for permafrost creep to occur".

Thank you for the suggestion. This phrase has been corrected!

---

## Author Response (AR3)

**Author comments (AC) in blue**

Dear authors,

thanks for submitting your revised version of the manuscript, addressing the referees' comments. This has greatly improved your manuscript! However, before the manuscript goes into production, I would like to have another round of revisions in which you pay close attention to the precision and conciseness of your language. Please review the manuscript carefully to eliminate unnecessary words, clarify ambiguous phrasing, and ensure that each sentence communicates its point as directly and effectively as possible. This final polish will help strengthen the impact and readability of your work.

Thank you for your kind feedback and for the opportunity to further improve our manuscript. Following your recommendation, we carefully reviewed the entire text with a focus on enhancing precision and conciseness. We have polished the language to eliminate unnecessary words, clarify any ambiguous phrasing, and ensure that each sentence communicates its point as clearly and directly as possible. We believe these final revisions have improved the overall readability and impact of the manuscript, and we hope it now meets the expectations for publication.

To give you a few examples where a clearer and more concise language is possible:

L205: The PSInSAR algorithm, as described by Rucci et al. (2012), Crosetto et al. (2016) and Poncoş et al. (2022), preserved --> The PSInSAR algorithm (references) preserved

It has been rephrased!

L210: An important number --> how many? Rather write, XXX out of YYY rock glaciers in the area...

It has been corrected! The number of rock glaciers was inserted.

L219: ... relatively homogeneous --> What does this mean in quantitative terms?

Unfortunately, RGIK (2023) does not currently provide a quantitative criterion for this classification. According to RGIK, a Moving Area (MA) is defined as *"an area at the surface of the rock glacier in which the observed direction and velocity of the flow field are spatially consistent and homogeneous during a documented timeframe."* Throughout their review, Reviewer 2 consistently emphasized the importance of adhering to the definition established by RGIK. To improve clarity and conciseness, we removed the word *"relatively."*

L233: had been surveyed ... --> were surveyed ...

It has been corrected!

L235: ... the roving receiver was moved to different points of interest in the field --> the roving receiver was moved in the field

It has been corrected!

L539: But rock glaciers experiencing very lower movement velocities --> But slow-moving rock glaciers were also documented in different ...

It has been corrected!

L550: ... revealed that, in many instances, the fronts of the rock glaciers --> revealed that the fronts of many rock glaciers ...

It has been corrected!

L556: The geophysical measurements performed in this study indicated the very low ice content --> Our geophysical measurements revealed a paucity of ice content

It has been corrected!

L566: ... the geophysical investigations presented in this paper --> ... our geophysical investigations reveal ...

It has been corrected!

L568: ... may be considerably lower ... --> use precise and concise language

It has been corrected!

There are currently countless examples were a more precise and concise wording would greatly enhance the readability of your text. Try to use quantitative information as much as possible while avoiding vague phrasing.

We have gone through the entire text and made corrections where we believed it was necessary.

And, once again, we appreciate your guidance in helping us strengthen the manuscript and believe these revisions have significantly improved its overall quality.

Kind regards,
Flavius Sîrbu

(on behalf of all co-authors)